# Reconstructing the in vivo dynamics of hematopoietic stem cells from telomere length distributions

**Benjamin Werner[1]\*[†], Fabian Beier[2][†], Sebastian Hummel[2], Stefan Balabanov[3], Lisa Lassay[4], Thorsten Orlikowsky[4], David Dingli[5,6], Tim H Brümmendorf[2], Arne Traulsen[1]\***

[1]Department of Evolutionary Theory, Max Planck Institute for Evolutionary Biology, Plön, Germany; [2]Department of Hematology and Oncology, Rheinisch-Westfälische Technische Hochschule Aachen University Hospital, Aachen, Germany; [3]Division of Hematology, University Hospital of Zürich, Zürich, Switzerland; [4]Department of Pediatrics, Rheinisch-Westfälische Technische Hochschule Aachen University Hospital, Aachen, Germany; [5]Division of Hematology, Department of Internal Medicine, Mayo Clinic, Rochester, United States; [6]Department of Molecular Medicine, Mayo Clinic, Rochester, United States

**Abstract** We investigate the in vivo patterns of stem cell divisions in the human hematopoietic system throughout life. In particular, we analyze the shape of telomere length distributions underlying stem cell behavior within individuals. Our mathematical model shows that these distributions contain a fingerprint of the progressive telomere loss and the fraction of symmetric cell proliferations. Our predictions are tested against measured telomere length distributions in humans across all ages, collected from lymphocyte and granulocyte sorted telomere length data of 356 healthy individuals, including 47 cord blood and 28 bone marrow samples. We find an increasing stem cell pool during childhood and adolescence and an approximately maintained stem cell population in adults. Furthermore, our method is able to detect individual differences from a single tissue sample, i.e. a single snapshot. Prospectively, this allows us to compare cell proliferation between individuals and identify abnormal stem cell dynamics, which affects the risk of stem cell related diseases.

**\*For correspondence:** werner@evolbio.mpg.de (BW); traulsen@evolbio.mpg.de (AT)

[†]These authors contributed equally to this work

**Competing interests:** The authors declare that no competing interests exist.

## Introduction

Homeostasis is in most mammalian tissues maintained by the occasional differentiation of infrequently dividing multi-potent stem cells (*Li and Clevers, 2010*; *Busch et al., 2015*). These cells are involved in the formation, maintenance, renewal, and aging of tissues (*Reya et al., 2001*; *Morrison and Kimble, 2006*). Their longevity imposes the risk of the accumulation of multiple mutations that potentially induce aberrant stem cell proliferation and can ultimately cause the emergence of cancer (*Hanahan and Weinberg, 2011*). The quantification of aberrant stem cell properties in cancer is impeded by the lack of detailed information about the expected patterns of cell replication in healthy human tissues (*Rossi et al., 2008*; *Vermeulen et al., 2013*). Dynamic properties of stem cell populations *in vivo* are predominantly obtained from sequential experiments in animal models (*Morrison and Spradling, 2008*; *Orford and Scadden, 2008*). Unfortunately, these methods are mostly inapplicable to humans and to infer *in vivo* properties of human stem cell populations remains a challenge. Indirect methods, i.e. biomarkers that reflect the proliferation history of a tissue, may overcome these limitations (*Greaves et al., 2006*; *Graham et al., 2011*; *Kozar et al., 2013*). In the

**eLife digest** Human cells die off regularly due to normal wear and tear, aging or injury. To replace these cells, humans maintain pockets of tissue specific stem cells that can develop into one of several different types of specialized cell. For example, stem cells in the bone marrow can develop into red blood cells, white blood cells or any of the other blood cell types. Unavoidably, over the course of a lifetime stem cells accumulate mutations that may cause them to become cancerous.

Researchers have learned a lot about stem cells by studying them under laboratory conditions. However, these studies cannot answer all the questions we have about human stem cells. As a result, human studies are needed; but frequently taking samples of stem cells from humans to assess them is impossible for numerous reasons, most importantly it is invasive and potentially harmful. Instead, researchers are looking for indirect ways to measure how stem cells grow.

Each time a cell divides, the protective ends of a chromosome – known as telomeres – get shorter. Now, Werner, Beier et al. have developed a mathematical model to assess human stem cell growth based on the length of the cells' telomeres. This model can gauge the growth patterns of the stem cell populations in an individual based on a sample taken from a single tissue.

Werner, Beier et al. tested the model using telomere measurements from blood and bone marrow samples taken from 356 healthy people of different ages. The results suggest that the stem cell population that gives rise to blood cells (the hematopoietic stem cells) increases in size during childhood and adolescence, but levels off during adulthood. The model also revealed that patterns of stem cell growth vary among individuals. Further studies of telomere length differences may help scientists identify the abnormal (stem cell-like) growth patterns associated with diseases like cancer.

following, we combine data of telomere length distributions and mathematical modelling of the underlying dynamical processes to deduce proliferation properties of human hematopoietic stem cells *in vivo*.

Telomeres are noncoding repetitive DNA sequences at the ends of all eukaryotic chromosomes. In vertebrates, these sequences consist of hundreds to thousands of repeats of the nucleobase blocks TTAGGG (*Griffith et al., 1999*). Telomere repeats are progressively lost in most somatic cells with age, as the conventional DNA polymerase is unable to fully copy the lagging DNA strand of chromosomes during cell replication (*Olovnikov, 1973*). Short telomeres are associated with genetic instability (*Hande, 1999*; *Feldser et al., 2003*). They trigger DNA-damage checkpoint pathways and enforce permanent cell cycle arrest ( *d'Adda di Fagagna et al., 2003*). Thus, telomere length limits the replication capacity of somatic cells (*Hayflick and Moorhead, 1961*) and can indirectly act as a tumor suppressor (*Kinzler and Vogelstein, 1997*; *Campisi, 2005*). This effect can be attenuated by the enzyme telomerase, which tags additional TTAGGG repeats to the end of chromosomes by utilizing single stranded RNA templates (*Greider and Blackburn, 1989*). Telomerase is primarily expressed in compartments of stem and germ line cells, as well as in numerous tumors (*Kim et al., 1994*). However, telomerase expression levels are insufficient to prevent the progressive loss of telomere repeats in most healthy human tissues with age (*Harley et al., 1990*; *Rufer et al., 1999*). This net loss of telomere repeats during cell replication leads to a characteristic telomere length distribution that reflects the replication history of cells. Since telomere length dynamics is important for a number of genetic and acquired disorders (*Hastie et al., 1990*; *Blasco, 2005*; *Calado and Young, 2009*), it is critical to understand the underlying mechanisms of this fundamental process. We have developed a mathematical model that allows us to interpret data of telomere length shortening in hematopoietic cells obtained from 356 healthy humans. Most importantly, we can infer the patterns of stem cell behavior from the underlying telomere dynamics within individuals from a single tissue sample, i.e. a single snapshot.

## Modelling telomere length dynamics

Our mathematical model recovers the temporal change of telomere length distributions in human hematopoietic cells with a minimal number of required model parameters. Since hematopoietic cells proliferate in a hierarchical organised tissue with slowly dividing stem cells at its root, such a model

needs to connect properties of cell proliferation and telomere shortening. Telomere length can be assessed on three different levels of resolution, (i) the level of single telomeres, (ii) the level of single cells and (iii) the level of the tissue. Of course these levels are not independent, for example the knowledge of telomere length in all cells allows to obtain the (average) telomere length of a tissue. The processes that drive telomere length dynamics differ at these levels of resolution. Single telomeres are prone to stochastic events such as oxidative stress or recombination and thus may also shorten by effects independent of proliferation associated attrition (*von Zglinicki, 2002*; *Antal et al., 2007*). Healthy human cells contain 184 telomeres, four on each of the 46 chromosomes. Thus, the noise on the level of single telomeres becomes much smaller on the cell level. We capitalise on this and consider telomere length on the cell level in the following. Thus, the average telomere length of a cell shortens by a constant factor during each division. Such an approach might underestimate the number of senescent cells once telomeres become critically short, since it is the length of the shortest telomere rather then the average telomere length that triggers cell cycle arrest (*Hemann et al., 2001*). Our model is sensitive to the accumulation of cells in the state of cell cycle arrest and we can infer this effect experimentally from population wide telomere length distributions. However, this effect can likely be neglected during adolescence and adulthood, but might have important implications in some tumors, at old age or in conditions associated with abnormal telomere maintenance.

We further need to consider properties of a hierarchical tissue organization, where few slowly dividing stem cells give rise to shorter lived progeny. Although some of the progeny, particularly primitive progenitor cells, can be long lived and are able to maintain homeostasis without stem cell turnover for intermediate time intervals, eventually all non hematopoietic stem cells will be depleted without continuous stem cell turn over (*Busch et al., 2015*; *Sun et al., 2014*). Age dependent differences in telomere shortening across different lineages of hematopoiesis can only persist in the hematopoietic system if they occur on the level of the maintained self-renewing cell population. Cells leaving the stem cell pool have an approximately constant number of cell divisions before they reach maturation (*Takano, 2004*; *Werner et al., 2011*). This shifts the distribution to shorter values of telomere length and consequently, the distribution of telomere lengths of mature cells is a good proxy for the distribution of telomere lengths in stem cells (*Rodriguez-Brenes et al., 2013*). We measured telomere length distributions in lymphocytes, granulocytes and bone marrow sections separately. This allows us to investigate the myeloid and lymphoid lineage of hematopoiesis independently.

In our model, we assume a population of initially $N_0$ stem cells. In the simplest case, each stem cell would proliferate with the same rate $r$ and the cell cycle time would follow an exponential distribution. However, tissue homeostasis requires continuous stem cell turn over in intermediate time intervals, therefore the proliferation rate of the population of stem cells is adjusted, such that a required constant output of differentiated cells per unit of time is maintained. In the simplest case of a constant stem cell population, the effective proliferation rate becomes $r/N_0$. However, in more complex scenarios, the number of stem cells could differ with age and the effective proliferation rate of stem cells $r/N(t)$ also becomes age dependent (*Rozhok and DeGregori, 2015*; *Bowie et al., 2006*). This resembles a feedback mechanism and results in an approximately Log-normal distribution of cell cycles, see also *Equation S26* in Materials and methods for details. In addition, each stem cell clone is characterised by a certain telomere length (*Antal et al., 2007*; *Simon and Derrida, 2008*). This telomere length shortens with each stem cell division by a constant length $\Delta c$ and consequently the remaining proliferation potential is reduced in both daughter cells (*Rufer et al., 1999*; *Allsopp et al., 1992*). If the telomeres of a cell reach a critically short length, this cell enters cell cycle arrest and stops proliferation, reflecting a cell's Hayflick limit (*Hayflick and Moorhead, 1961*). This can be modelled by collecting cells with the same proliferation potential in states $i$. A cell enters the next downstream state $i \rightarrow i+1$ after a cell division, see also *Figure 1*, as well as *Equations S1,S14* in Materials and methods. Since the next cell to proliferate is chosen at random from the reservoir, cells progressively distribute over all accessible states with time (*Olofsson and Kimmel, 1999*). This corresponds to the problem of how many cells are expected in a state $i$ at any given time, which we denote by $N^{(i)}(t)$ in the following.

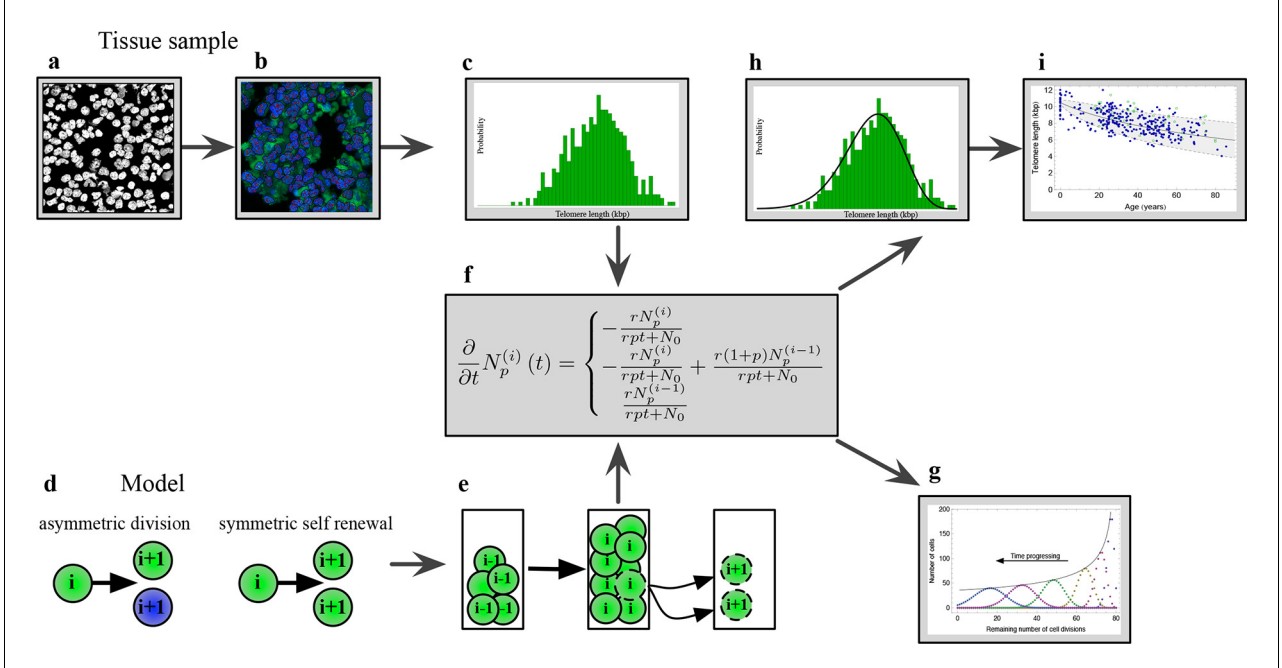

**Figure 1.** The combination of telomere length data and mathematical modeling allows to infer individualized stem cell proliferation patterns. (a–c) Blood or bone marrow samples were taken from healthy persons with ages between 0 and 85. Telomere length was measured with Flow-FISH and Q-FISH techniques, resulting in individualized telomere length distributions. (d–g) Mathematical framework: Stem cells divide either symmetrically or asymmetrically. Each cell is characterized by an average telomere length. Cells with the same state are collected in compartments. The average of the underlying stochastic process is captured by a system of differential equations. The solution of this equation is a generalised truncated Poisson distribution that gives rise to a traveling wave, see *Equation S15*. (h, i) The combination of modeling and telomere length distribution measurements allows dynamic predictions for individuals, see Figure 6. These predictions can be tested on population wide data of telomere length, for example see Figure 3.

The following figure supplement is available for figure 1:

**Figure supplement 1.** Results of the mathematical model on the temporal change of individual telomere length distributions.

## Results

### The model predicts characteristic telomere length distributions for different ratios of symmetric and asymmetric stem cell divisions

The shape of the distribution of cells across cell cycles depends on the patterns of stem cell proliferation, for example the ratio of symmetric versus asymmetric divisions. An asymmetric stem cell division produces one stem and one non-stem cell (for example a progenitor cell that leaves the stem cell compartment). If we restrict the stem cells dynamics to only asymmetric divisions, the process results in a stem cell population of constant size and the number of cells in each state $i$ follows a Poisson distribution

$$N^{(i)}(t) = \frac{N_0}{i!}\left(\frac{rt}{N_0}\right)^i e^{-\frac{rt}{N_0}}. \tag{1}$$

A typical example of this distribution is shown in *Figure 1—figure supplement 1* and details on the derivation can be found in Materials and methods, see *Equation S1*. Cells with maximum proliferation capacity (cells in state 0 in our model) are progressively lost and cells accumulate in the final state of cell cycle arrest by passing through all intermediate states.

Inferring the dynamics of distribution (1) from in vivo measurements requires sequential sampling and complicated cell sorting, which seems challenging in realistic clinical settings. On the other hand, the measured (observed) telomere length distribution corresponds to a single sample of the

underlying Poisson process. The expected shape of this observed distribution is depicted in *Figure 1g*. It becomes a traveling wave that starts narrowly distributed around an initial telomere length and shifts towards shorter average telomere length with time. We have measured this distribution, which arises from our theoretical model, experimentally in many samples of granulocytes, lymphocytes and bone marrow sections of healthy adult humans, which we discuss in detail below.

In addition to asymmetric divisions, stem cells can undergo symmetric self-renewal, which is a prerequisite for development, as it allows for a growing stem cell population. In our model, stem cells divide symmetrically with probability $p$ and asymmetrically with probability $1-p$ respectively. In this situation, the number of stem cells is not constant, but increases with each symmetric stem cell self-renewal. As a consequence, the expected distribution also changes and is now described by a generalised Poisson distribution (see *Equation S14* in Materials and methods) given by

$$N_p^{(i)}(t) = \frac{N_0}{i!} \left(\frac{1+p}{p}\right)^i \frac{\ln^i\left(\frac{rp}{N_0}t + 1\right)}{\sqrt[p]{\frac{rp}{N_0}t + 1}}.$$ (2)

This distribution also leads to a traveling wave, but the maximum of the distribution decreases considerably slower compared to the case of purely asymmetric stem cell divisions. In the following, we refer to the model that is restricted to only asymmetric stem cell divisions as model 1 and denote the more general case of symmetric and asymmetric cell divisions as model 2.

Ideally, we would like to follow these traveling waves in individual healthy humans over time and compare this sequential data to the dynamics from our model predictions. Unfortunately, the time required to confirm our model across all ages would exceed the life expectancy of the authors. We therefore explored those properties of our analytical model that are directly testable in population wide data of telomere length. One such property is the change of the average telomere length with age, which we measure in a group of 356 healthy individuals.

## The average telomere length decreases nonlinearly in the presence of symmetric stem cell self-renewal

The average telomere length decreases in most human tissues with age (*Harley et al., 1990*). This is well known and has been confirmed numerous times. Surprisingly, less is known about the detailed dynamics of this decrease. We can derive the dynamics of the average telomere length from the telomere length distributions directly. The average telomere length corresponds to the expected value of the telomere length distribution (in the following denoted by $E[c(t)]$), see *Equation S5* in Materials and methods for details. As the telomere length distribution changes with time, the average telomere length becomes time dependent naturally. In the absence of symmetric stem cell self-renewal (model 1) the average telomere length $E[c(t)]$ is expected to decrease linearly

$$E[c(t)] \approx c - \Delta c \frac{rt}{N_0},$$ (3)

with age (denoted by $t$ in the equation above). More specifically, the average telomere length of cells of a particular type, e.g. the population of granulocytes or lymphocytes, shorten by a constant fraction each year. The dynamics changes once a significant fraction of cells enter cell cycle arrest, see *Equation S9*. The average telomere length transitions from a linear into a power law decline (when the average telomere length becomes very short) and the stem cell pool reaches the state of complete cell cycle exhaustion asymptotically. This transition would enable the identification of an age where a considerable fraction of stem cells enter cell cycle arrest, potentially a mechanism important in aging, carcinogenesis or bone marrow failure syndromes.

Furthermore, we calculated the variance of the underlying stochastic process. This gives us a measure for the expected fluctuation of the average telomere length in a population of healthy humans. We expect the variance to increase linearly in time in the absence of symmetric stem cell self-renewal. Consequently, the standard deviation is proportional to the square root of age. Yet again, similar to the average telomere length, the dynamics of the variance changes once a significant fraction of cells enters cell cycle arrest. The variance starts to decrease and would reach zero, if all cells stopped proliferation.

The distribution of telomere length changes under the presence of symmetric stem cell self-renewal (model 2). Accordingly, we expect a different decrease of the average telomere length. We find that the telomere length follows a logarithmic decay with age (see also *Equation S19*), given by

$$E_p[c(t)] \approx c - \Delta c \frac{1+p}{p} \ln\left(\frac{rp}{N_0}t + 1\right)$$

(4)

The average telomere length of a cell population shortens less with increasing age under the presence of symmetric self-renewal, although the decrease of telomeric repeats per cell division (denoted by $\Delta c$ in *Equation 4*) is constant. This effect emerges naturally in our model due to the increasing number of stem cells with age. In a population with only few cells, each cell proliferation has a considerable impact on the average telomere length, while this impact diminishes in larger populations. If the stem cell population increases progressively, telomere shortening reduces on the tissue level with age.

## In vivo measurements of telomere length suggest an increasing number of hematopietic stem cells during human adolescence

In order to test the predictions of our model experimentally, we have measured telomere length in lymphocytes and granulocytes in a cohort of 356 healthy humans with ages between 0 and 85 years. Our data includes 47 cord blood samples of healthy children and bone marrow biopsies of 28 patients with diagnosed Hodgkin lymphoma without bone marrow involvement. We assessed the average telomere length in all 356 samples with established Flow-FISH protocols (*Aubert et al., 2012*; *Baerlocher et al., 2006*; *Weidner et al., 2014*; *Beier et al., 2012*). This reveals the population wide dynamics of telomere length and contains a significant number of cord blood samples that allow us to investigate differences in cell proliferation during adolescence and homeostasis in adulthood.

In addition, we have analyzed 28 blood samples of lymphocytes, 10 blood samples of granulocytes and 28 bone marrow biopsies with quantitative-fluorescence in situ hybridisation (Q-FISH) (*Beier et al., 2015*; *Varela et al., 2011*; *Zijlmans et al., 1997*) (see *Figure 2* and experimental methods for details). The averages of these samples correspond to the open symbols in *Figure 3*. From the full distribution, we obtain the telomere length distributions of single individuals and estimate personalised cell proliferation properties, e.g. the ratio of symmetric to asymmetric cell divisions as well as the rate of telomere shortening for each sample separately. We compare these personalised estimates to population wide telomere length to test the consistency of our results on two independent data sets.

In order to compare our model with the experimental data, we implemented standard maximum likelihood estimates for a regression analysis. Our experimental finding in adults (we only consider persons of 20 years or older) show that telomere length in granulocytes and lymphocytes decreases approximately linearly with age on the population level. In both cell populations the telomere length of adults decreases with $50 \pm 5 \text{ bp/year}$ (we state the maximum likelihood estimate and the 95% confidence interval). If for example a cell looses on average 50 bp telomeric repeats per cell division (*Rufer et al., 1999*), this implies approximately 1 replication per year for the hematopoietic stem cells. This agrees with the observation of rare stem cell turnover under homeostasis (*Busch et al., 2015*; *Sun et al., 2014*; *Dingli et al., 2006*).

However, the assumption of strictly asymmetric cell divisions (model 1) fails to explain the pronounced loss of telomere repeats in infants (prediction of model 1 for the initial telomere length in lymphocytes: $9.8 \pm 0.15 \text{ kbp}$, measured average initial telomere length: $10.67 \pm 0.4 \text{ kbp}$, similar results for granulocytes, see also *Figure 4* for a comparison of model 1 and model 2). This discrepancy can be resolved by introducing an interplay of symmetric and asymmetric stem cell divisions (model 2) that allows for an increasing number of stem cells. In this situation, the proliferation rate of stem cells becomes age dependent and our model predicts that at the youngest ages, when the number of stem cells is lowest, telomere loss is most pronounced. Maximum likelihood estimates of our general mathematical solution (*Equation 4*) to the telomere length data on the population level (see *Figure 3*) reveals for the parameter controlling average loss of telomere length in lymphocytes a value of $75 \pm 7 \text{ bp/year}$, an initial telomere length of $10.4 \pm 0.2 \text{ kbp}$ and a probability for symmetric stem cell self-renewal of $0.35 \pm 0.07$. In granulocytes we find a value of telomere loss of $68 \pm 5 \text{ bp/year}$, an initial telomere length of $10.2 \pm 0.3 \text{ kbp}$ and a probability for symmetric stem

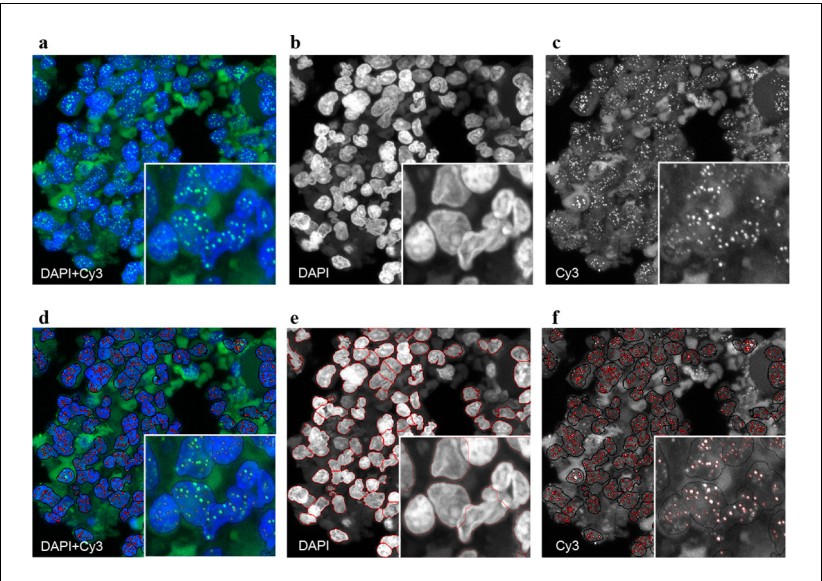

**Figure 2.** Representative image of the Q-FISH analysis of a bone marrow section. (**a**) Maximum projection image of a paraffin-embedded bone marrow section of confocal Q-FISH with DAPI and Cy3. (**b, c**) Single DAPI and Cy3 staining respectively. (**d**) Overlay of image analysis of nucleus and telomere detection. (**e**) Image analysis of the DAPI staining is shown. Detected nuclei are shown in red. (**f**) Image analysis of the Cy3 staining. Detected telomeres marked in red. For details on the Q-FISH analysis please see Materials and methods.

The following figure supplements are available for figure 2:

**Figure supplement 1.** Representative image of the Q-FISH analysis of a peripheral blood cytospin.

**Figure supplement 2.** Representative FACS blot of a flow-FISH analysis.

cell self-renewal of $0.44 \pm 0.2$. This probability accounts for the increased loss of telomere repeats in infants and substantially improves the prediction of the initial average telomere length. In addition to our group of 356 healthy humans, we have tested our hypothesis in an independent data set of 835 healthy humans, previously published by an unrelated group in (*Aubert et al., 2012*), see *Figure 3—figure supplement 1*. This set confirms our parameter estimations, in particular the accelerated decrease of average telomere length during adolescence is also observed.

Our model suggests that the increased loss of telomere repeats in the first years of human life is a consequence of an expanding stem cell population. This expansion is combined with a reduction in proliferation rates of single stem cells. The loss of telomere repeats during cell replication has a more pronounced impact on the average telomere length within a small cell population and diminishes in large stem cell populations. This explains the increased loss of telomeric repeats during adolescence (see *Figure 4*) naturally as a consequence of growth by an expanding stem cell population. Similarly, a sudden accelerated loss of telomeric repeats in aged individuals could point towards an insufficient stem cell self-renewal. This might provide a promising direction for further investigations with an extended data set of sufficiently high resolution in aged individuals.

## Proliferation properties of stem cells differ during adolescence and adulthood

Our analytical model is consistent with population wide telomere length data. It shows that symmetric stem cell self-renewals are more frequent in adolescence and their effect on the dynamics of average telomere length reduces with age. However, how robust are our conclusions under variation of model parameters or a change of cell proliferation properties with age? One possibility to address these problems is the implementation of Bayesian inference methods (*Dempster, 1968*). In a nutshell, such methods draw a random set of model parameters either from an uninformed (objective)

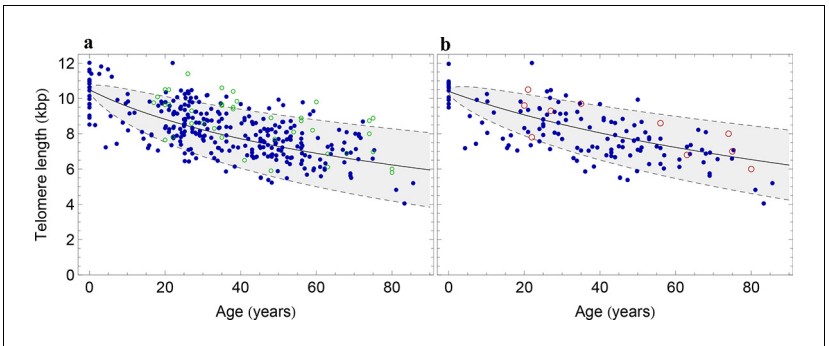

**Figure 3.** The population wide average telomere length of (**a**) lymphocytes and (**b**) granulocytes. The data from a cohort of 356 individuals (symbols) is captured by a logarithmic decrease of the average telomere length (solid line), which is predicted by our model 2 that allows for symmetric stem cell divisions and thus leads to a slowly increasing stem cell pool. Based on the fit of the average, the mathematical model predicts a standard deviation that increases with the square root of the age (dashed lines). This approach does not take the genetic variability of telomere length in newborns into account. The decrease of the average telomere length slows down in children and becomes almost linear in adults, see also Figure 4. For individuals represented by filled symbols, only information on the average telomere length is available. For individuals represented by open symbols, we additionally analysed the distribution of individually detected telomeres, see Figure 6. An additional parameter estimation on an independent data set is shown in *Figure 3—figure supplement 1*.

The following figure supplement is available for figure 3:

**Figure supplement 1.** Decrease of the average telomere length of (**a**) lymphocytes and (**b**) granulocytes in a population of 835 healthy humans.

or informed (subjective) prior distribution and produce independent realizations of the model. These realizations are compared to some (appropriate) data of interest and fits with a predefined statistic significance are retained while unsatisfactory realizations are rejected. Originally developed for phylogenetic tree reconstruction, such methods are increasingly used in other applications (*Marjoram and Tavaré, 2006*). Bayesian inference methods allow to quantify the uncertainty in an analysis by providing posterior distributions of model parameters.

In the following we implement an Approximate Bayesian Computation (ABC) rejection sampling framework (*Csilléry et al., 2010*) on the data presented in *Figure 3*. We derive posterior distributions for our three free model parameters, the initial telomere length $c$, the relative decrease of telomere length per time $\Delta cr/N_0$ and the probability of symmetric stem cell divisions $p$. We draw these variables independently from uniform (uninformed) distributions and test $10^9$ independent realizations of our mathematical model 1 and model 2. We seek parameter regimes that maximize the coefficient of determination $R^2$ between *Equation 3* (model 1) or *Equation 4* (model 2) and the average telomere length presented in *Figure 3*. We discard any parameter combination below a threshold. We perform the same analysis independently on the data set of granulocytes and lymphocytes.

In both cases, we find localized posterior parameter distributions. For lymphocytes, parameters peak at $\Delta cr/N_0 = 0.071 \pm 0.005$ kbp/year, $c = 10.41 \pm 0.3$ kbp and $p = 0.32 \pm 0.2$, see *Figure 5c–e*. Only a small parameter range explains the exact patterns of telomere shortening. We find approximately 70% of stem cell divisions are asymmetric and 30% are symmetric self-renewals. This stochastic approach confirms the results of the non-linear model fits using a standard maximum likelihood approach that were discussed in the previous section, but provides further information on the distribution of our parameters.

The previous analysis assumes a fixed set of parameters for the dynamics of telomere shortening for all ages. In principle, these parameters could also change with age. To see if we can identify ages with different stem cell proliferation parameters, we investigated a third model that allows for successive phases of stem cell dynamics with independent parameter sets for each phase. We consider an additional parameter $t_T$, which corresponds to a transition time. We perform the above Bayesian

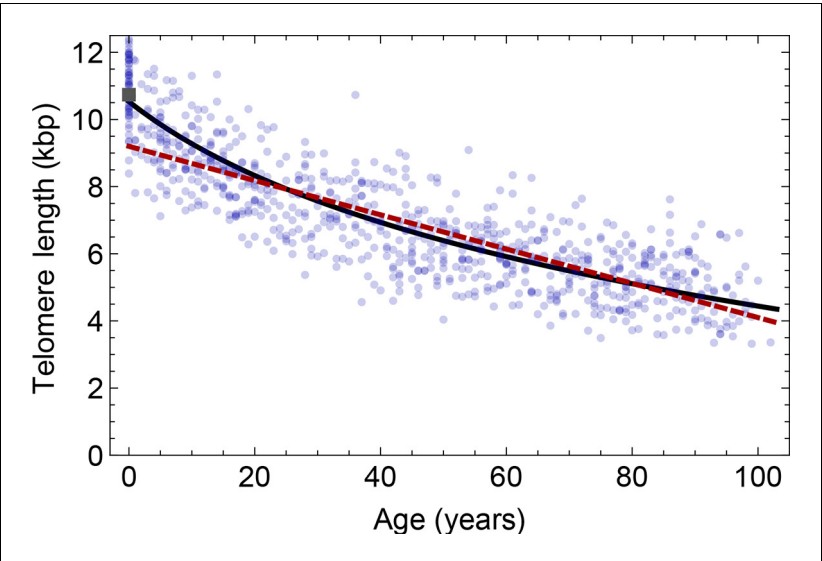

**Figure 4.** Comparison of the average telomere length decrease of lymphocytes predicted by Model 1 and Model 2. Model 1 (red dashed line, best fit to the data) predicts a linear decrease of the average telomere length with age. The linear decrease underestimates the initial accelerated telomere loss during adolescence (the average initial telomere length in newborns is shown by the dark grey rectangle). In contrast, model 2 (black line) predicts a logarithmic decay of the average telomere length with age and is able to capture the increased loss of telomere length during adolescence, as well as the approximately linear decrease in adults.

approach independently for each random partition of the data set. This approach suggests at most two separate phases, with a transition between the 6th and 7th year of life for lymphocytes, see *Figure 5f–i*, and a transition between the 10th and 15th years of life for granulocytes, see *Figure 5j–m*. In infants and the first years of life, the probability of stem cell self-renewal shows a significant variance (*Figure 5*). However, the data resolution is insufficient for this short time window to provide reliable parameter estimates. The probability of symmetric stem cell self-renewal in adults however is in the range of $p \in (0, 0.2)$. This is lower as was predicted by the regression analysis across all ages. This suggests a reduction in the self-renewal probability of stem cells after adolescence and points towards an either slower growing or constant stem cell population in adults. This may reflect selection for an optimal stem cell population size to minimize the risk of cancer initiation as suggested in theoretical studies before (*Michor et al., 2003*).

Next, we aimed to test which of the three models explains the data best, considering the complexity of the models. We therefore utilise the likelihood estimates of the former subsection and perform a model selection based on the Akaike information criterion (AIC) (*Burnham, 2004*). Model 1 scores with an AIC of 2550, model 2 has an AIC of 2328 and a multiphase model with a minimum of 7 parameters yields an AIC of 2361. The AIC is minimized by model 2. Based on this approach, model 1 as well as a multiphase model can be rejected as more likely explanations for the telomere length shortening presented in *Figure 3* (given the above numbers and according to standard procedures, the relative likelihood of model 1 to better explain the data compared to model 2 is assumed to be $p \approx 10^{-48}$, the relative likelihood of the multi-phase model to better explain the data compared to model 2 is assumed to be $p \approx 10^{-8}$). This selection is robust under the choice of different statistical methods. For example, a BIC approach selects the models in the same order.

## A single sample of the telomere length distribution can inform about stem cell dynamics

The actual stem cell population sizes and their dynamics do not only vary with age, but also between individuals. This has immediate consequences on the susceptibility of individuals towards certain diseases (*Calado and Young, 2009*; *Brümmendorf and Balabanov, 2006*) and could potentially be used in individualised treatment strategies. Our model describes the telomere length distributions in

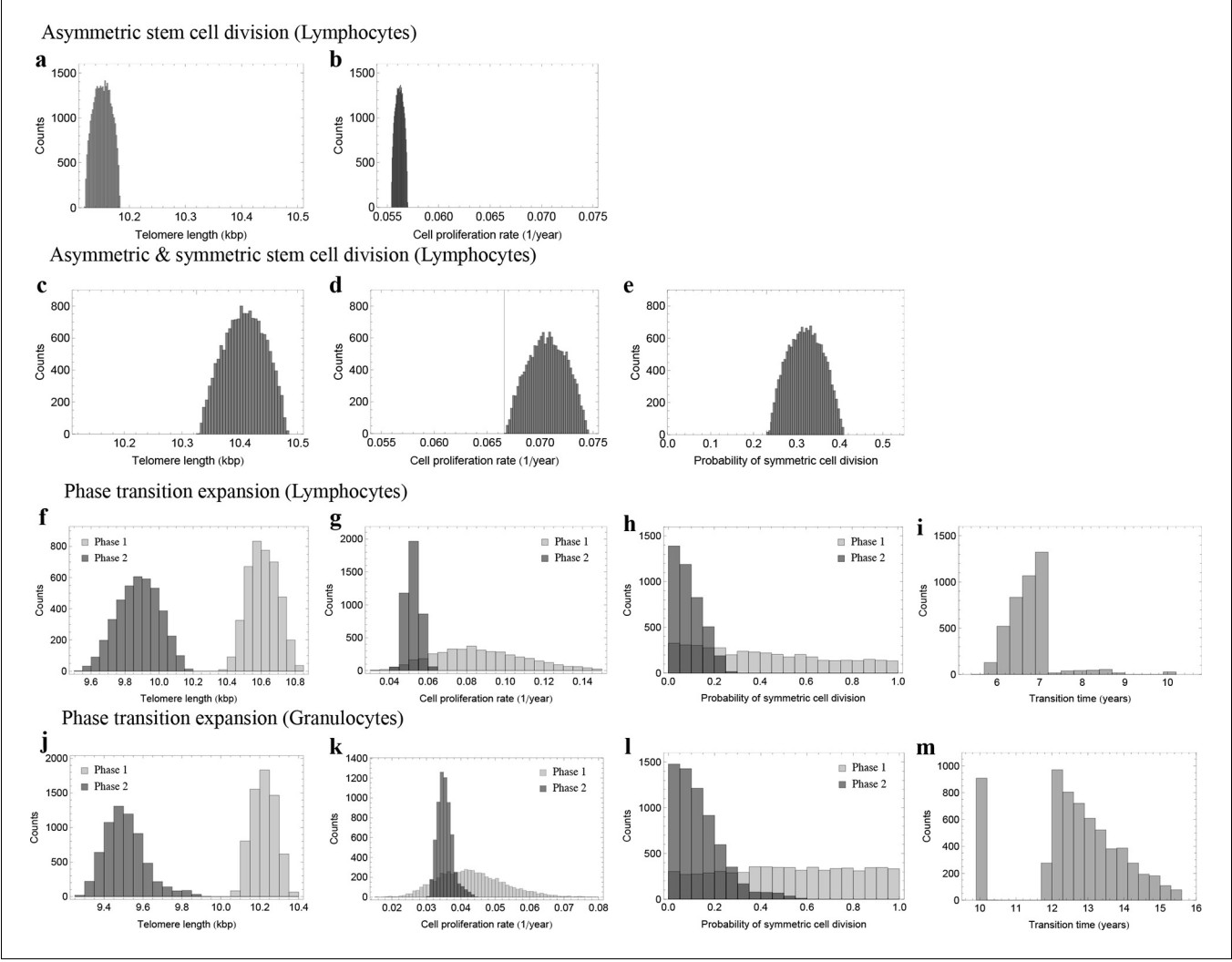

**Figure 5.** Posterior distributions of model parameters from Approximate Bayesian Computation (ABC). (**a, b**) Model fit for only asymmetric stem cell divisions (model 1) to the data of average telomere length on the population level. The expected telomere length decreases linearly and two free model parameters, i.e. initial telomere length and stem cell turn over rate are estimated. (**c–e**) ABC with symmetric and asymmetric stem cell divisions (model 2). In this case one additional free parameter (probability of symmetric stem cell divisions) can be estimated. (**f–i**) ABC for a two phase extension of the model inferred from population wide data of lymphocytes, panels (**j–m**) show the same analysis for granulocytes. A likelihood based model selection favours model 2 and rejects model 1 as well as the multiphase model as more likely explanations for the observed data.

individuals and quantifies three parameters, i.e. initial telomere length, increase of stem cell pool size and stem cell replication rates of an individual from a single tissue sample. We therefore extended our experimental protocols to further test our theoretical results. First, we measured single telomere signals of peripheral blood sorted for lymphocytes in 28 individuals and sorted for granulocytes in 10 individuals by quantitative confocal FISH in addition to the average telomere length that is provided by flow FISH. Second, we investigated the telomere length distribution in paraffin-embedded bone marrow sections of an additional cohort of 28 healthy individuals using quantitative confocal FISH (*Beier, 2005*), see *Figure 2*. We compare our general telomere length distribution that allows for any ratio of symmetric and asymmetric stem cell divisions (model 2) to the data set of all 66 individuals. Cases of four representative individuals are shown in *Figure 6*. All cases can be found in *Figure 6—figure supplements 1–3* and all individual cell proliferation properties as well as quality of fits are summarised in *Supplementary file 1*. The average telomere length of these 66 distributions are shown as open symbols in *Figure 3*.

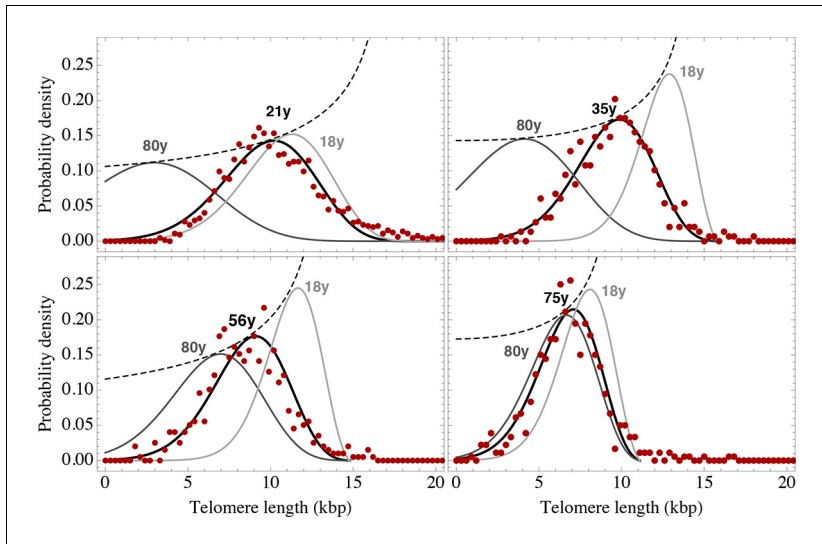

**Figure 6.** Telomere length distributions of granulocytes for four representative individuals. Telomere length distributions within the nucleus of individual cells are measured once in single individuals (symbols). This data is fitted with our model 2 (black line, see *Equation S29* for details), leading to estimates for the parameters of the theoretical distribution. These parameters can be used to extrapolate the distribution to any other age (gray lines). The dashed line shows the prediction for the maximum of the distribution (*Equation S18*). Telomere length distributions differ between individuals and change in different patterns, depending on the exact proliferation parameters in individuals. Additional cases are shown the *Figure 6—figure supplements 1–3*. A summary of all fitting parameters can be found in *Supplementary file 1*.

The following figure supplements are available for figure 6:

**Figure supplement 1.** Nonlinear fits of the expected telomere length distribution to telomere length distributions of granulocytes in peripheral blood of 10 healthy donors.

**Figure supplement 2.** Nonlinear fits of the expected telomere length distribution to telomere length distributions of lymphocytes in peripheral blood of 28 healthy donors.

**Figure supplement 3.** Nonlinear fits of the expected telomere length distribution to telomere length distributions in bone marrow biopsies of 28 patients with diagnosed M. Hodgkin without bone marrow affection.

The fits of our calculated distribution (see *Equation S15* for the distribution and *Equation S29* for details on the fitting procedure) reveal substantial differences in initial telomere length, increase of stem cell pool size and stem cell replication rates between the 66 individuals, but also between granulocytes, lymphocytes and bone marrow samples. We find a low probability of symmetric self-renewal ($p$ between 0.005 to 0.03 per cell division) in all individual samples. This agrees with our results on the average telomere length shortening in adults at the population level and supports our observation of an approximately maintained active stem cell number in individuals after adolescence. Also the average telomere loss per year varies between individuals and ranges from 18 bp/year to 110 bp/year. However, the averages of all individual parameter sets agree with the estimated proliferation properties inferred from the population wide data of telomere length. We find differences between individual samples of lymphocytes and granulocytes. While the loss of telomeric repeats slows down with age in granulocytes, it slightly accelerates in lymphocytes, see *Figure 7*. These cells represent the myeloid and lymphoid lineage respectively. In our model, such a reduced rate of telomere loss can be explained with an increased reservoir of myeloid specific stem and progenitor cells and is in agreement with a skewed differentiation potential towards the myeloid lineage of aged hematopoietic stem cells (*Geiger et al., 2013*).

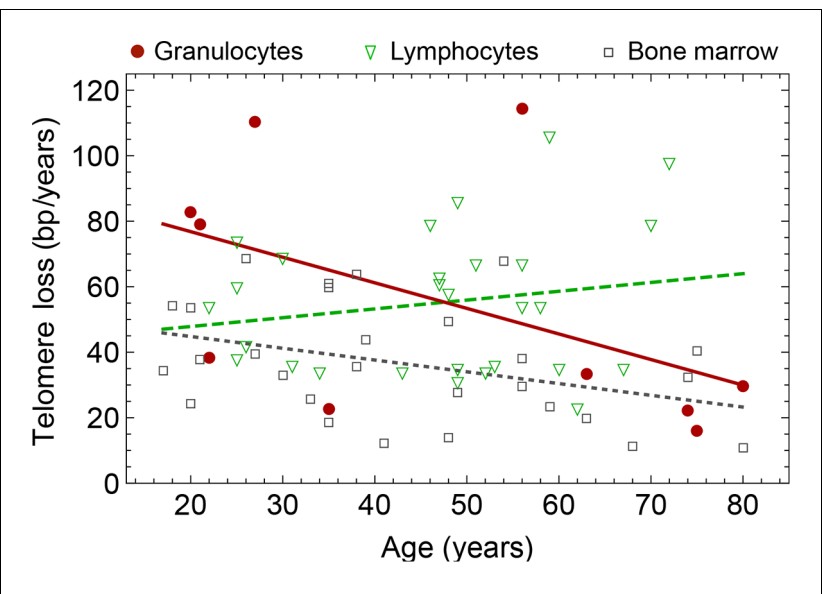

**Figure 7.** Rate of telomere loss in 66 individuals. Shown is the rate of telomeric shortening (bp/year) of granulocytes (circles), lymphocytes (triangle) and bone marrow sections (rectangle), inferred from telomere length distributions of 66 different individuals (see *Figure 5* and supplemental figures and *Supplementary file 1* for a summary of all parameters). Differences between individuals are large, but the average telomere shortening rate conforms to parameter estimates of population wide data of telomere length, see for example *Figure 5*. Cells in the bone marrow show a lower proliferation rate and consequently the rate of telomere loss is reduced (gray dotted line). The rate of telomere loss decreases with age in granulocytes (−0.78 bp/year, dark red line) and in bone marrow sections (−0.36 bp/year, grey dotted line), but increases in lymphocytes (+0.27 bp/year, dark green dashed line). This observation agrees with a skewed differentiation potential towards the myeloid lineage of aged hematopoietic stem cells (*Geiger et al., 2013*). The lines are only meant to represent a trend of increase or decrease with age. The change with age is most probably not linear.

## Discussion

Our knowledge about the dynamics of tissue specific stem cells comes mostly from lineage tracing experiments in transgenic mouse models. They provided insights into many aspects of tissue formation and maintenance, e.g. the intestinal crypt, but also the hematopoietic system (*Busch et al., 2015*; *Sun et al., 2014*; *Itzkovitz et al., 2012*). However, there is variation between different transgenic mouse models and their significance for human stem cell properties remains a challenging question. In some cases, clonal lineages can be traced by naturally occurring somatic mutations, e.g. particular mtDNA mutations in human intestinal crypts (*Baker et al., 2014*). However, the *in vivo* dynamic properties of human hematopoietic stem cells remain poorly characterized.

Here, we have utilized telomere length distributions of hematopoietic cells as a biomarker that contains information about the proliferation history of cells. We developed a mathematical model that allows us to infer dynamic properties of stem cell populations from data of telomere length distributions. These properties were analyzed in different cell types, e.g. lymphocytes, granulocytes and bone marrow sections of individuals of different ages. These calculated distributions describe the change of telomere length within the human population. The expected changes with age were confirmed in a representative group of 356 healthy individuals and the conclusions are consistent with our individualized parameter estimations.

The population wide data of average telomere length reveals different stem cell properties in adolescence and adulthood. Telomere length decrease is logarithmic and occurs at a faster rate during adolescence, suggesting a stem cell pool expansion in the first years of human life compatible with growth. This decrease becomes almost linear in adults and is in line with an approximately constant stem cell population. It is an interesting question why the number of stem cells would reach a certain targeted size. This could be simply because of spatial constrains in the bone marrow. Yet, from an evolutionary perspective, intermediate stem cell pool sizes were suggested to minimize the

risk of cancer initiation (*Rodriguez-Brenes et al., 2013*; *Michor et al., 2003*). Such an optimization requires feedback signals that ensures the maintenance of an intermediate sized stem cell population, feedback signals that might be prone to (epi)genetic change and potentially are involved in cancer and ageing.

It is still a debated question if stem cells in mammals are maintained by predominantly asymmetric divisions, or alternatively by a population strategy of balanced symmetric self-renewal and symmetric differentiation. While the former strategy can be implemented on the single cell level, the latter strategy would require further feedback signals. From a modelling perspective, a population strategy of symmetric self-renewal and symmetric differentiation was suggested to minimize the clonal load within a stem cell population (*Shahriyari et al., 2013*). On the other hand, experimental findings seem to point towards predominantly asymmetric divisions, but this might also differ across tissues (*Morrison and Kimble, 2006*). In our model, the stem cell pool is maintained by asymmetric cell divisions. A balance of symmetric and asymmetric cell divisions would on average result in the same telomere length dynamics and thus would be indistinguishable from asymmetric divisions on the population level, only the interpretation of $p$, the probability of symmetric self-renewal would change in this case. Yet, the variance of the distribution would be expected to increase under the presence of symmetric differentiation and symmetric self-renewal. However, likely this effect is weak compared to the measurement related noise of telomere length.

Our method quantifies the parameters of telomere dynamics from a single blood sample or paraffin-embedded tissue samples of an individual. It is independent of any particular tissue organization and thus can be applied, in principle, to any tissue. This general method will be of particular interest to distinguish stem cell dynamics in healthy and sick individuals. We expect characteristic changes in telomere length distributions in certain (hematopoietic) stem cell disorders such as chronic leukemias (*Braig et al., 2014*) and bone marrow failure syndromes (*Calado and Young, 2009*; *Beier, 2005*). Therefore, our model can serve as a tool to infer stem cell dynamics *in vivo* retrospectively and prospectively from a single tissue sample. Such an approach can not only increase our understanding of disease dynamics but may also contribute to personalized disease diagnosis and prognosis in the future.

## Materials and methods

### Patients

Peripheral blood of 309 healthy blood donors was obtained from the blood donor bank in Aachen. Q-FISH of peripheral blood cytospins was performed on 28 healthy blood samples. 47 cord blood and blood samples from healthy children and adolescents were obtained from the Department of Pediatrics and Neonatology of the University Hospital of Aachen. Bone marrow biopsies of 28 patients with diagnosed Hodgkin lymphoma without bone marrow involvement were used for bone marrow analysis. All samples were taken with informed consent and according to the guidelines of the ethics committees at University Hospital Aachen.

### Flow-FISH

The Flow-FISH technique provides the mean telomere length per nucleus. Flow-FISH was carried out according to previously published protocols (*Aubert et al., 2012*; *Baerlocher et al., 2006*; *Weidner et al., 2014*; *Beier et al., 2012*). Briefly, after osmotic lysis of erythrocytes with ammonium chloride, white blood cells were mixed with cow thymocytes. Cells were hybridized with FITC labeled, telomere specific (CCCTAA)3- peptide nucleic acid (PNA) probe (Panagene) and DNA was counterstained with LDS 751 (Sigma). FACS analysis was carried out on Navios or FC-500 (both Beckman Coulter). Thymocytes, lymphocytes and granulocytes subsets were identified based on LDS571 staining and forward scatter. Mean telomere length was calculated by subtracting the unstained autofluorescence value of the respective lymphocyte, granulocyte or thymocyte subpopulation. Cow thymocytes with a determined telomere length were used as an internal control to convert telomere length in kilobase (kb). All measurements were carried out in triplicate.

## Quantitative-Fluorescence in situ hybdridsation (Q-FISH)

Q-FISH offers the possibility to analyze the distribution pattern of individual telomeres. For cytospins of peripheral blood cells, erythrocytes were lysed using ammonium chloride (Stem cell Technologies, Vancouver, British Columbia, Canada) and 50,000 cells were centrifuged for cytospin. Cells were fixed with 70% ethanol solution for 30 seconds and air dried for 15 min. Bone marrow sections were deparaffinized with xylol and rehydrated with ethanol following standard protocols. Deparaffinized bone marrow tissue sections, metaphases and peripheral blood cells were processed following previously published protocols (*Beier et al., 2015*; *Varela et al., 2011*; *Zijlmans et al., 1997*). After initial washing with PBS, slides were fixed in formaldehyde (Sigma) (4%) in PBS for 2 min. Slides were further washed (three times for 5 min) with PBS followed by dehydration with ethanol and air drying for 30 min. Hybridization mixture containing 70% formamide (Sigma), 0.5% Magnesium chloride (Sigma), 0.25% (wt/vol) blocking reagent (Boeringer) 0.3 µg/ml Cy-3-conjugated (C3TA2)3 peptide nucleic acid probe (Pnagene), in 10 mM Tris (pH 7.2, Sigma) was added to the slide. After adding a coverslip; DNA was denatured for 3 min at 85°C. Hybridization was carried out for 2 h at room temperature. After washing the slides twice with 70% formamide/10 mM Tris (pH 7.2)/0.1% bovine serum albumin (BSA), slides were washed again (three times for 5 min) with 0.05 M Tris/0.15 M NaCl (pH 7.5) containing 0.05% Tween-20. After dehydration with ethanol slides were air dried and stained with PBS containing 0.1 ng/ml of 4'-6-diamidino-2-phenylindole (DAPI) for 5 min. After mounting the cells (Vectashield, Vectorlabs), a coverslip was added.

## Image analysis

Confocal microscopy analysis was carried out at a Leica TCS-sp5 confocal microscope (Leica). Images were acquired at 63x magnification and 1.5-2.0 digital zoom. Multi-tracking mode was used to acquire images. Stacks of DAPI and Cy3 staining were taken with a step size of 1 µm. Peripheral blood cells and bone marrows were captured including five steps (z-range 4 µm). Maximum projection of the images was carried out and Definiens XD 1.5 image analysis software (Definiens GmbH) was used for quantitative image analysis. Nucleus and telomere detection was carried out based on DAPI and Cy3 intensity patterns. A valid image analysis was assumed in case of a correct detection of 90% of all visible telomeres. All image analysis was carried out single-blinded. Individual telomere signals were calculated after subtraction of the mean background value per detected nucleus. For bone marrow section and peripheral blood cells, values of all detected telomeres were used for analysis. Paraffin embedded lymphocytes of three healthy donors and granulocytes of a patient with chronic myeloid leukemia with a determined telomere length were used as controls for bone marrow biopsies. Linear regression of the control cells was carried out to convert telomere length from arbitrary units to kb. Telomere length in kb of the Q-FISH analysis of peripheral blood cells was calculated based on the linear regression of the corresponding Flow-FISH values.

## Mathematical model of telomere length dynamics

We assume a finite number of $1 + c$ accessible telomere states of stem cells, where each state $i$ contains cells of equal average telomere length. Initially, $N_0$ cells are in state $0$ and cells will progressively enter downstream states after cell divisions. An asymmetric division of a cell in state $i$ leads to one more differentiated cell (more committed within a hierarchically tissue organization) and one stem cell. The committed (progenitor) cell leaves the pool of stem cells and does not further contribute to dynamics in the stem cell population. The second cell keeps the stem cell properties and enters state $i + 1$, reflecting the shortening of its telomeres by a length of $\Delta c$. Similarly, a symmetric cell division results in two stem cells, both entering the next subsequent state. In our model, stem cells divide symmetrically with probability $p$ and asymmetrically with probability $1 - p$, respectively. A cell in state $c$ enters cell cycle arrest and cannot reach subsequent states - the next proliferating cell is randomly chosen amongst all cells not yet in state $c$.

### Stochastic simulations

We implement individual based stochastic simulations of our telomere model. We initialize our program with $N_0$ cells in state $0$. The next cell to proliferate is chosen randomly amongst all cells not yet in state $c$. If a cell is chosen, we draw a random number $\xi \in [0, 1]$. If $\xi > p$, one cell enters the next subsequent compartment (corresponding to an asymmetric cell division). If $\xi \leq p$, two cells enter the

next subsequent compartment (corresponding to a symmetric stem cell division). In both cases, the mother cell is removed. Iterating over many cell divisions leads to a distribution of cells amongst the accessible $1 + c$ cell cycle states. Recording the temporal change of the distribution allows us to infer further properties of interest such as the time dependence of the average and the variance of the distribution. All simulations are implemented in *C++*, and are analyzed and visualized in *Mathematica 10.0* and *R 3.2.1*.

## Asymmetric cell divisions

We first discuss the telomere length dynamics under asymmetric cell divisions (corresponding to $p = 0$ and called model 1 in our further notation). We call $N^{(i)}(t)$ the number of cells in state $i$ at time $t$. We further choose the initial condition $N^{(0)}(0) = N_0$. Asymmetric cell divisions strictly conserve the size of the cell pool $\sum_{i=0}^{c} N^{(i)}(t) = N_0$. We apply a deterministic, time continuous approximation of the underlying stochastic process and capture the average dynamics of telomere shortening by a system of coupled differential equations,

$$\dot{N}^{(i)}(t) = \begin{cases} -r\dfrac{N^{(i)}}{N_0} & i = 0 \\[2ex] -r\dfrac{N^{(i)}}{N_0} + r\dfrac{N^{(i-1)}}{N_0} & 0 < i < c \\[2ex] r\dfrac{N^{(i-1)}}{N_0} & i = c. \end{cases} \tag{S1}$$

Here, $r$ represents the proliferation rate of a cell. Cells move towards higher states progressively and accumulate in state $c$, where they enter cell cycle arrest.

The general solution of (*Equation S1*) can be derived recursively and is given by

$$N^{(i)}(t) = \begin{cases} \dfrac{N_0}{i!}\left(\dfrac{rt}{N_0}\right)^i e^{-\frac{rt}{N_0}} & 0 \leq i < c \\[3ex] N_0\left(1 - \displaystyle\sum_{l=0}^{c-1}\dfrac{1}{l!}\left(\dfrac{rt}{N_0}\right)^l e^{-\frac{rt}{N_0}}\right) & i = c. \end{cases} \tag{S2}$$

The number of cells in states $i < c$ resembles a truncated Poisson distribution with rate parameter $\frac{r}{N_0}$ and shape parameter $i$. *Figure 1g* shows a comparison of solution (*Equation S2*) to exact individual based stochastic computer simulations. The number of cells in state $0$ decreases exponentially. Cells in states $i = 1, \ldots, c - 1$ are initially absent, undergo a maximum and vanish in the long run again. Only cells in state $c$ accumulate over time.

Inferring distribution (*Equation S2*) from *in vivo* data requires several blood samples at sequential time intervals. A single measurement of the telomere length distribution at time $t'$ corresponds to the interception points of a vertical line, drawn at time $t'$, and the number of cells in every state in the model given by *Equation S2*. Thus, the observed distribution at time $t'$ in *Figure 1g* is given by

$$f_{t'}(c) = \{N^{(0)}\left(t'\right), \ldots, N^{(c)}\left(t'\right)\}. \tag{S3}$$

This distribution becomes a traveling wave that shifts towards shorter average telomere length in time, see *Figure 1—figure supplement 1*. The maximum of this wave reaches state $i$ after time $t_{\max}^{(i)} = \frac{iN_0}{r}$. Plugging this into *Equation S2*, we find for the maximum of this traveling wave

$$N^{(i)}\left(t_{\max}^{(i)}\right) = \frac{N_0}{i!}\left(\frac{i}{e}\right)^i \approx \frac{N_0}{\sqrt{2\pi i}} = \frac{N_0}{\sqrt{\frac{2\pi r}{N_0}t_{\max}^{(i)}}}, \tag{S4}$$

where we applied Stirling's formula. The most abundant telomere length declines proportional to $\frac{1}{\sqrt{t_{max}}}$ in time if cells undergo asymmetric cell divisions only.

Next we calculate the time dependence of the average telomere length $E[c(t)]$. This corresponds to the first moment of the distribution (*Equation S2*), given by

$$
\begin{aligned}
E[c(t)] &= \frac{1}{N_0} \sum_{i=0}^{c} (c - \Delta c i) N^{(i)}(t) \\
&= \sum_{i=0}^{c} \frac{c - \Delta c i}{i!} \left( \frac{rt}{N_0} \right)^i e^{-\frac{rt}{N_0}} \\
&= \sum_{i=0}^{c} \frac{c}{i!} \left( \frac{rt}{N_0} \right)^i e^{-\frac{rt}{N_0}} - \Delta c \sum_{i=0}^{c} \frac{i}{i!} \left( \frac{rt}{N_0} \right)^i e^{-\frac{rt}{N_0}},
\end{aligned}
\tag{S5}
$$

where cells in state $c$ do not contribute. To calculate this sum we first note that the upper incomplete gamma function is defined as $\Gamma\left[1 + c, \frac{rt}{N}\right] = \int_{\frac{rt}{N}}^{\infty} dx \; x^c e^{-x}$, but can also be represented by incomplete exponential sums $\Gamma\left[1 + c, \frac{rt}{N}\right] = c! e^{-\frac{rt}{N}} \sum_{i=0}^{c} \frac{1}{i!} \left( \frac{rt}{N} \right)^i$. If we set $x = \frac{rt}{N}$, we can write

$$
\sum_{i=0}^{c} \frac{c}{i!} x^i e^{-x} = \frac{c}{c!} \Gamma[1 + c, x]
\tag{S6}
$$

the second term is

$$
\sum_{i=0}^{c} \frac{i}{i!} x^i e^{-x} = x \sum_{i=0}^{c} \frac{x^i}{i!} e^{-x} + x \frac{\partial}{\partial x} \sum_{i=0}^{c} \frac{x^i}{i!} e^{-x},
\tag{S7}
$$

and thus we have

$$
e^{-x} \sum_{i=0}^{c} \frac{i}{i!} x^i = x \sum_{i=0}^{c} \frac{x^i}{i!} e^{-x} + x \frac{\partial}{\partial x} \sum_{i=0}^{c} \frac{x^i}{i!} e^{-x} = x \frac{\Gamma[1 + c, x]}{c!} - \frac{x^{1+c}}{c!} e^{-x}.
\tag{S8}
$$

In the last step we used the property of the upper incomplete gamma function $\frac{\partial}{\partial x} \Gamma[n + 1, x] = -x^n e^{-x}$. Collecting all terms in *Equation S5* again gives

$$
E[c(t)] = \frac{\Delta c}{c!} \left( \frac{rt}{N_0} \right)^{1+c} e^{-\frac{rt}{N_0}} + \frac{cN_0 - \Delta c rt}{N_0} \frac{\Gamma\left[1 + c, \frac{rt}{N_0}\right]}{c!}.
\tag{S9}
$$

The expression for the average telomere length (*Equation S9*) simplifies significantly for certain parameter regimes. For example for the hematopoietic system in humans we expect $N_0$ at least to be in the order of a few hundred of cells and $c$ is strictly larger than zero. Thus the first term in *Equation S9* is very small and negligible. The second term is dominated by the linearly decaying term, as the incomplete gamma function is $\Gamma\left[1 + c, \frac{rt}{N_0}\right] \approx c!$ for $t \ll r/N_0$, i.e. sufficiently small $t$. Thus in this situation expression (S9) is well approximated by

$$
E[c(t)] \approx \frac{cN_0 - \Delta c rt}{N_0}
\tag{S10}
$$

until only few cells have reached state $c$. The linear approximation *Equation S10* is excellent, until most cells reach states of very short telomeres. In the situation of critically short telomeres, the full solution (*Equation S9*) has to be used and the average telomere length reaches zero asymptotically.

Our approach allows us to calculate additional properties of the system. The knowledge of the exact distribution enables us to derive all moments of the distribution. For example, we can derive analytical expressions for the time dependence of the variance $\sigma^2(t)$. First note, that the moment generating function for the distribution (*Equation S2*), $M_c(z) = E[e^{cz}](t)$, is

$$M_c(z) = 1 + \frac{e^{(e^{-z}-1)\frac{rt}{N_0}}\Gamma\left[1+c, \frac{e^{-z}rt}{N_0}\right]}{c!} - \frac{\Gamma\left[1+c, \frac{rt}{N_0}\right]}{c!}. \tag{S11}$$

We recover the average (*Equation S9*) of the telomere length distribution via $E[c(t)] = \frac{\partial}{\partial x}(M_c(0))$. The variance can be calculated via

$$\sigma^2(t) = E\left[c(t)^2\right] - E^2[c(t)] = \frac{\partial^2}{\partial x^2}M_c(0) - \left(\frac{\partial}{\partial x}M_c(0)\right)^2$$

$$= \left(\frac{rt}{N_0}\right)^{1+c}\frac{N_0 c - rt}{N_0 \ c!}e^{-\frac{rt}{N_0}} + \left[\left(c - \frac{rt}{N_0}\right)^2 + \frac{rt}{N_0}\right]\frac{\Gamma\left[1+c, \frac{rt}{N_0}\right]}{c!} - E^2[c(t)]. \tag{S12}$$

Again, the first term of *Equation S12* is negligible for a biological meaningful parameter range. The quadratic term $(c - rt/N_0)^2$ is compensated by an identical term in $E^2[c(t)]$ (see *Equation S9*). Again, the gamma function is approximately equal to $c!$ for sufficiently small times. Thus, expression (*Equation S12*) is initially dominated by the linear term and consequently, the variance grows linear as $\sigma^2 = \frac{rt}{N_0}$. The standard deviation increases in time as

$$\sigma = \sqrt{\frac{rt}{N_0}}. \tag{S13}$$

The linear approximation of the variance is excellent. Only if cells start to accumulate in state $c$ (cell cycle arrest) the variance decreases.

## Symmetric cell divisions

In the following, we modify the system of differential *Equation S1* (model 1) to incorporate symmetric stem cell divisions (model 2). We assume a cell division to be symmetric with probability $p$ and asymmetric with probability $1 - p$ respectively. Note that the number of stem cells is not constant but increases due to symmetric cell divisions. Initially there are $N_0$ cells with telomeres of length $c$. We assume a number of stem cell divisions that is constant within a fixed time interval, reflecting the necessity to produce a fixed number of differentiated cells during a unit of time. However, time intervals between stem cell divisions remain stochastic in the individual based model. As a consequence, the stem cell pool increases linearly in time, $N_p(t) = N_0 + rpt$. Thus, the system of differential equations changes to

$$\dot{N}_p^{(i)}(t) = \begin{cases} -\dfrac{rN_p^{(i)}}{rpt + N_0} & i = 0 \\[3mm] -\dfrac{rN_p^{(i)}}{rpt + N_0} + \dfrac{r(1+p)N_p^{(i-1)}}{rpt + N_0} & 0 < i < c \\[3mm] \dfrac{rN_p^{(i-1)}}{rpt + N_0} & i = c. \end{cases} \tag{S14}$$

The solution to this system of differential equations is

$$N_p^{(i)}(t) = \begin{cases} \dfrac{N_0}{i!}\left(\dfrac{1+p}{p}\right)^i \sqrt[p]{t^*}\ln^i(t^*) & 0 \le i < c \\[4mm] N_0(1+p)^{i-1}\left(1 - \dfrac{\Gamma(i, p^{-1}\ln(t^*))}{(i-1)!}\right) & i = c \end{cases} \tag{S15}$$

where we used $t^* = \frac{rp}{N_0}t + 1$ as an abbreviation. Using l'Hopital and $e^x = \lim_{n\to\infty}\left(1 + \frac{x}{n}\right)^n$ we recover the *Equation S2* for $p \to 0$ and the solution turns into a Poisson distribution again,

$$\lim_{p \to 0} N_p^{(i)}(t) = \frac{N_0}{i!}\left(\frac{rt}{N_0}\right)^i e^{-\frac{rt}{N_0}} = N^{(i)}(t). \tag{S16}$$

Note that we assumed a constant number of cell divisions within a fixed time interval. Due to the increasing stem cell pool size, this effectively causes a reduction in the proliferation rate of individual stem cells with age.

Similar to the former subsection, the time dependence of the maximum of the distribution can be calculated for $i = 1, \ldots, c-1$. The time until the maximum of the telomere length distribution reaches length $i$ becomes

$$t_{\max,p}^{(i)} = N_0 \frac{e^{ip}-1}{rp}. \tag{S17}$$

The time to reach the maximum increases exponentially in $i$ for symmetric cell divisions, in contrast to the linear increase for only asymmetric cell divisions. However, *Equation S17* reduces to the result we obtained in the former subsection in the limit $p \to 0$. The cell count at the maximum becomes

$$N^{(i)}\left(t_{\max,p}^{(i)}\right) \approx \frac{N_0(1+p)^i}{\sqrt{2\pi i}}. \tag{S18}$$

The maximum decreases considerably slower with $i$ (given the same initial size of the stem cell pool) compared to the case of only asymmetric cell divisions *Equation S4*, where we have used Stirling's formula for the approximation. Similar to the former subsection we can calculate the average of the telomere length distribution. This time the average becomes

$$\begin{aligned} E_p[c(t)] &= \frac{1}{N_p(t)} \sum_{i=0}^{c} (c - i\Delta c) N_p^{(i)}(t) \\ &= \frac{\Delta c \rho^{1+c}}{c!} \frac{\ln^{1+c}(t^*)}{(t^*)^\rho} + \frac{\Gamma\left[1+c, \rho \ln(t^*)\right]}{c!}(c - \Delta c \rho \ln(t^*)) \end{aligned} \tag{S19}$$

with $t^* = \frac{rp}{N_0}t + 1$ and $\rho = \frac{1+p}{p}$. Similar to *Equation S9*, this expression is dominated by the second term of the equation. The average decreases approximately logarithmically for sufficiently small $t$,

$$E_p[c(t)] \approx c - \Delta c \frac{1+p}{p} \ln\left(\frac{rp}{N_0}t + 1\right). \tag{S20}$$

The temporal decrease of the average telomere length speeds up with decreasing $p$. In the limit $p \to 0$, we recover the result (*Equation S10*) of a linear decreasing average. Similar to the former section we can derive the variance of the distribution, using the moment generating function $M_p(x) = E_p[e^{cx}](t)$, via

$$\sigma_p^2(t) = \frac{\partial^2}{\partial x^2} M_p(0) - \left(\frac{\partial}{\partial x} M_p(0)\right)^2. \tag{S21}$$

However, the result becomes less accessible and informative. Thus we restrict ourselves to a numerical solution of *Equation S21*. The logarithmic decay of the average telomere length has consequences on the interpretation of experimental results of telomere length distributions. In infants an accelerated decrease of telomere length can be observed. This can be explained immediately by an expanding stem cell pool. The stem cell pool contains only a few $N_0$ stem cells initially (newborns). These stem cells divide symmetrically with probability $p$ and asymmetrically with probability $1-p$ respectively. The symmetric cell divisions cause an increase of the stem cell pool size and an indirect decrease in cell proliferation rates. The logarithmic decay is pronounced initially, but flattens after some time (as the number of stem cells increases). Thus, in adults the logarithmic decay is difficult to distinguish from a linear decay, see for example *Figure 4* in the main text.

## Connections to the Normal and Log-Normal distribution

The number of cells in each state $i$ follows a Poisson distribution

$$N^{(i)}(t) = \frac{N_0}{i!}\left(\frac{rt}{N_0}\right)^i e^{-\frac{rt}{N_0}} \tag{S22}$$

in the case of only asymmetric stem cell divisions, see *Equation S2* for details. We introduce $x = \frac{rt}{N_0}$, and upon normalisation (S22) becomes

$$N^{(i)}(x) \propto \frac{x^i}{i!}e^{-x}, \tag{S23}$$

where $x$ is a Poisson distributed variable. For $x$ sufficiently large, this random variable is well described by a normal distribution and we have $x \propto$ Normal distribution.

If we allow for symmetric cell divisions, cells in state $i$ followed a generalised Poisson distribution

$$N_p^{(i)}(t) = \frac{N_0}{i!}\left(\frac{1+p}{p}\right)^i \ln^i\left(\frac{rpt}{N_0}+1\right) \sqrt[-p]{\frac{rpt}{N_0}+1}, \tag{S24}$$

see *Equation S15* for details. Choosing $y = \frac{rpt}{N_0}+1$ and neglecting normalisation factors we can write

$$N_p^{(i)}(y) \propto \frac{1}{i!}\ln^i(y) \sqrt[p]{y} \tag{S25}$$

If we change variables again and choose $y = e^x$, *Equation S25* becomes

$$N_p^{(i)}(y=e^x) \propto \frac{1}{i!}\ln^i(e^x) \sqrt[p]{e^x} = \frac{x^i}{i!}e^{-\frac{x}{p}} \propto N^{(i)}(x). \tag{S26}$$

As $x = \frac{rt}{N_0}$ is approximately normally distributed, and $y = e^x$, $y = \frac{rpt}{N_0}+1$ follows a Log-normal distribution.

## Parameter evaluation for the average telomere length on population level by Bayesian inference method

We implement Approximate Bayesian Computation (ABC) rejection samplings to derive posterior parameter distributions for the predicted average telomere length under asymmetric (model 1, *Equation S10*) and combined symmetric and asymmetric (model 2, *Equation S20*) cell proliferations respectively. Utilizing *Equation S10*, we have to infer two parameters: (i) the average decrease of telomere length per time $r/N_0$ and (ii) the initial telomere length $c$. In the case of *Equation S20* a third variable has to be determined: (iii) the probability of symmetric cell divisions $p$. We draw these variables independently from uniform distributions (prior) with ranges $r/No \in [0,0.2]\frac{\text{kbp}}{\text{year}}$, $c \in [7,15]$ kbp and $p \in [0,1]$ and produce $5 \times 10^8$ independent realizations of *Equation S10,S20*. We calculate the coefficient of determination $R^2$ between each of these realizations and the average telomere length from a data set of 356 healthy individuals (see for example *Figure 1* in the main text) via

$$R^2 = 1 - \frac{\sum_i(E[c](t_i) - y(t_i))}{\sum_i(\overline{y} - y(t_i))}. \tag{S27}$$

Here, $y(t_i)$ denotes, the measured telomere length of an individual with age $t_i$, $\overline{y}$ is the average measured telomere length of the population and $E[c](t_i)$ the value of a single realization of *Equation S10* or *Equation S20* at time $t_i$ given the random set of parameter values. We seek parameter regimes that maximize $R^2$ and discard any parameter combination below a certain threshold.

### Bayesian parameter evaluation for asymmetric cell divisions

For a linear fit according to *Equation S10* with 2 parameters we find $R^2_{\max} = 0.5314$ as the maximum value for the coefficient of determination. To determine the possible rate of parameters we discard any parameter combination with $R^2 < 0.53$. This gives sharp posterior distributions for both parameter values that peak at $\Delta cr/N_0 = 0.056 \frac{\text{kbp}}{\text{year}}$ and $c = 10.15$ kbp, see *Figure 5a,b*. This concurs with best parameter estimations from linear fitting $c_f = 9.85 \pm 0.2$ kbp and $\Delta cr_f/N_f = 0.05 \pm 0.005 \frac{\text{kbp}}{\text{year}}$.

This scenario underestimates the initial telomere length ($c = 10.15$, whereas the average initial telomere length in the data is $\bar{c} = 10.67$ kbp).

## Bayesian parameter evaluation for an interplay of symmetric and asymmetric cell divisions

For a logarithmic fit according to *Equation S20* with three parameters we get an improved coefficient of determination $R^2_{\max} = 0.541$. We discard any parameter combination that results in $R^2 < 0.54$. Again we find localized posterior parameter distributions that peak at $\Delta cr/n_0 = 0.071 \frac{\text{kbp}}{\text{year}}$, $c = 10.41$ kbp and $p = 0.32$, see *Figure 5c–e*. This approach improves the prediction of the initial telomere length. The average loss of telomere length per year is higher compared to only asymmetric proliferation and the probability of symmetric cell divisions peaks in a range of $p \in [0.25, 0.4]$. This concurs with a nonlinear fit, where we find $p_f = 0.37 \pm 0.2$, $c_f = 10.4 \pm 0.3$ kbp and $\Delta cr_f/N_f = 0.071 \pm 0.005 \frac{\text{kbp}}{\text{year}}$. However, we note this is an average over all individuals with an age distribution from $0$ to $85$.

## Bayesian parameter evaluation for a phase transition extension of the model

In the following we partition the data into two subsets and analyze an extension of the model. We introduce an additional parameter $t_T$ that resembles a transition time. This transition time is drawn from a uniform distribution with $t_T \in [0, 80]$. We perform above Bayesian approach according to *Equation S20* independently for each random partition of the data set. This gives in total seven posterior distributions. This approach gives $R^2_{\max} = 0.573$ as the maximum value for the coefficient of determination and we discard any parameter combination with $R^2 < 0.57$. The transition occurs in children at the age of 6 to 7, see *Figure 5f–i*, and a clear distinction of the posterior parameter distributions between phase 1 and phase 2 can be observed. The parameter estimations confirm with the interpretation of a growing stem cell pool. We find an increased rate of telomere shortening, compared to phase 2 as well as an increased probability of symmetric cell divisions.

## Non linear fitting of calculated telomere length distributions to measured distributions in single individuals

In the previous subsection, the average telomere shortening at the population level was investigated. We found indications for an increasing stem cell pool with age in particular in children due to infrequent symmetric stem cell divisions. In the following, we shift from the population level towards the telomere length distribution in healthy individuals. *Equation S15* allows us to compare theoretical predictions to measured telomere length distributions and to infer individual proliferation parameters of stem cell populations in vivo from a single blood sample under an interplay of symmetric cell divisions (with probability $p$) and asymmetric cell divisions (with probability $1 - p$). However, *Equation S2* is contained as the special case ($p = 0$), according to *Equation S15*. The expected number of cells that have not entered cell cycle arrest is given by

$$N_p^{(i)}(t) = \frac{N_0}{i!}\left(\frac{1+p}{p}\right)^i \frac{\ln^i\left(\frac{rp}{N_0}t+1\right)}{\sqrt[p]{\frac{rp}{N_0}t+1}}. \tag{S28}$$

We set $t^* = \frac{rp}{N_0}t + 1$, normalize (S28) and obtain for the expected telomere length distribution

$$\rho_p(x,t) = \frac{1}{(c-x)!(t^*)}\left(\frac{1+p}{p}\right)^{c-x}\sqrt[-p]{t^*}\ln^{c-x}(t^*). \tag{S29}$$

We perform non-linear fits of *Equation S29* to measured telomere distributions in healthy individuals, leaving three free parameters $t^*$, $p$ and $c$ to be determined. Results of the nonlinear fits can be seen in *Figure 6—figure supplements 1–3*. The corresponding fitting parameters are denoted in *Supplementary file 1*.

## Acknowledgements

We would like to thank Lucia Vankann for technical assistance. Confocal microscopy was performed in the "Immunohistochemistry and confocal microscopy" core unit of the Interdisciplinary Center for Clinical Research (IZKF) Aachen within the Faculty of Medicine at RWTH Aachen University with support of Gerhard Müller-Newen.

## Additional information

### Funding

No external funding was received for this work.

### Author contributions

BW, Approved the manuscript, Conception and design, Analysis and interpretation of data, Drafting or revising the article; FB, Approved the manuscript, Acquisition of data, Analysis and interpretation of data, Drafting or revising the article; SH, Approved the manuscript, Acquisition of data; SB, DD, Approved the manuscript, Drafting or revising the article; LL, TO, Approved the manuscript, Drafting or revising the article, Contributed unpublished essential data or reagents; THB, AT, Approved the manuscript, Conception and design, Drafting or revising the article

### Ethics

Human subjects: All samples and the approval for publication were taken with informed consent of all patients at the University Hospital Aachen according to the guidelines and the approval of the ethics committees at the University Hospital Aachen.

## Additional files

### Supplementary files

• Supplementary file 1. (A) Best parameters from fitting the calculated distribution S19 to telomere length distributions of granulocytes from 10 adult persons (see *Figure 6—figure supplement 1*). Here $p$ denotes the probability that a stem cell proliferation results in two additional stem cells, $c$ is the initial telomere length in kbp and $-\Delta cr/N_0$ corresponds to the loss of telomere repeats in bp/year. (B) Best parameters from fitting the calculated distribution S19 to telomere length distributions of lymphocytes from 28 adult persons (see *Figure 6—figure supplement 2*). (C) Best parameters from fitting the calculated distribution S19 to telomere length distributions of bone marrow samples from 28 adult persons (see *Figure 6—figure supplement 3*).

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
