## [Decision Letter]

Thank you for submitting your work entitled "Reconstructing the in vivo dynamics of hematopoietic stem cells from telomere length distributions" for peer review at *eLife*. Your submission has been favorably evaluated by Naama Barkai (Senior Editor) and three reviewers, one of whom is am member of our Board of Reviewing Editors.

The reviewers have discussed the reviews and the Reviewing Editor has drafted this decision to help you prepare a revised submission. The referees find the work very interesting and of high quality. Using simple models for symmetric and asymmetric stem cell divisions, key parameter that characterizes the in vivo dynamics of telomere length distributions are reconstructed from measured telomere length distributions.

Before the manuscript could be accepted, several points raised by the referees should be addressed and clarified in a revised manuscript. These concern in particular the possibility to use a likelihood based inference approach and questions about the relationship between measured data (telomere length) and model states (numbers of stem cell divisions).

Reviewer #1:

The authors study the length distributions of telomeres in hematopoietic stem cells. Using simple stochastic models of telomere shortening during cell division, parameters characterizing the in vivo dynamics of hematopoietic stem cells are reconstructed.

This work is elegant and very interesting. Overall the paper is based on a simple and elegant idea. It provides information on stem cell dynamics in humans. However, when reading the manuscript it remains unclear to me how robust and sound the conclusions and results are. There are a number of points that need to be addressed by the authors before the strength of the work can be fully appreciated.

Major points:

A basic problem I have when reading the manuscript is that the models of telomere length reduction address the telomere length distribution in the stem cell population. However the telomere length data is obtained for the cell population taken from blood or bone marrow that contains differentiated cells. So in order to analyze the data it appears that one would need a model for the telomere length distribution of those cells found in the samples. Why should this distribution correspond to the one of the stem cell pool described by the model? This is quite unclear.

A related point is that the authors discuss very simple scenarios where only one step of differentiation from the stem cell pool is considered. In practice, differentiated blood cells could arise via several differentiation steps in a more complex scheme. This might change the telomere length distribution of the circulating cells. Such issues are not addressed in the paper.

A question arises from the statement in that "a cell loses typically 30 to 50kbp telomeric repeats per cell division" (subsection “In vivo measurements of telomere length suggest an increasing number of hematopietic stem cells during human adolescence”). This implies that the loss of repeats per division is itself a stochastic variable with mean and variance. In this case the distribution of telomere length differs from the distribution of cell states *i*. This is not discussed. Rather, the distinction is blurred by a confusing mix of language. For example in the subsection “Proliferation properties of stem cells differ during adolescence and adulthood”, the units of *r/N_0_* are given as kbp/year even though by definition *r/N_0_* is a rate with units 1 over time.

The presentation of the work is sometimes confusing because of imprecise language and confusing terminology. For example, in the subsection “The model predicts characteristic telomere length distributions for different ratios of symmetric and asymmetric stem cell divisions” you state: "Each stem cell proliferates randomly with a rate *r*". Again, in the subsection “Asymmetric cell divisions”: "*r* represents the proliferation rate of a cell". However looking carefully at Equation (S1) suggests that the parameter *r* is not what the authors say it is. The proliferation rate per cell rather appears to be given by the ratio *r/N_0_* when using the terminology of the authors. Similarly, for the case with symmetric divisions, the proliferation rate per cell is *r/(rpt+N_0_)*. This is consistent with the statement that the proliferation rate of a stem cell decreases with time but is inconsistent with the definition of *r* in the text.

The statement "the assumption of strictly asymmetric divisions fails to explain the pronounced loss of telomere in infants" is unclear. What exactly is the evidence for a pronounced loss in infants? If one fits a linear law to the data in Figure 2 there would be a lot of variability at young age but not a "pronounced loss".

Related to this last point is the statement in the caption to Figure 2: "The decrease of the average telomere length slows down in children and becomes almost linear in adults." I do not understand how this statement follows from the data shown. The data does not appear to support this statement.

The statement of fact that the "increased loss" of telomere length at young age is "an immediate consequence of an expanding stem cell population" is a strong statement, without a clear reasoning. It rather seems to be a suggestion based on the model that is here stated as plain fact. Also, the evidence for an "increased loss" is not compelling, as it is not easily seen in the data in Figure 2 do not understand where this claim stems from. The statement about "increased loss… during adolescence" is unclear too. Which part of the data is meant here?

The distinction of two phases discussed in Figure 4 is not very convincing and seems a bit arbitrary. Also, in Figure 4, it is unclear what data was analyzed with the Bayesian method. What problem is solved by introducing the distinction between two phases? The Bayesian approach also becomes conceptually puzzling when the probability p is now itself becoming distributed by a very broad probability distribution in phase 1 (Figure 4). What does this mean?

The linear fits to the data in Figure 6 are not really compelling as the data seems to be consistent with widely varying linear behaviors.

Reviewer #2:

In their manuscript entitled 'Reconstructing the in vivo dynamics of hematopoietic stem cells from telomere length distributions' Werner et al. use a simple mathematical model of symmetric/asymmetric stem cell divisions and link it telomere length distributions measured experimentally in hematopoietic cells. Studying the properties of the model, they identify qualitative differences in the resulting telomere distribution between a strictly asymmetric and a mixed symmetric/asymmetric model of cell division. Fitting the model to the data, they claim that during childhood stem cell divisions are predominately symmetric leading to an expanding stem cell pool, while during adulthood stem cell divisions are mostly asymmetric.

The manuscript is well written, the theoretical results/calculations are well explained, sometimes maybe even in too much detail in the Introduction but definitely sound, and the application of the model to experimental data reveals interesting insight into (individual) human stem cell dynamics.

I very much liked that the authors took a Bayesian approach to model estimation, because the indeterminacies in the model need to be quantified. My main criticism (details below) however revolves around some modeling aspects and the precise inference method used to fit the model to the experimental data.

Before publication, I encourage the authors include the following improvements/clarification into the manuscript:

Major points:

The applied inference approach (termed 'Bayesian inference method') seems to be very simplistic. The authors essentially apply Approximate Bayesian Computation (ABC) rejection sampling (this term should be mentioned in the manuscript). These likelihood-free methods are usually applied when the likelihood function of the model is intractable. However, the authors show (e.g. S10, S20) that they can solve for the relevant quantities of their model. Hence a likelihood based inference approach is possible (e.g. maximum likelihood estimation + profile likelihoods, or full MCMC for posterior distributions) and should be used (in order to avoid the pitfalls associated with ABC). In fact, fitting S10 is just a linear regression problem.

You consider three different models of increasing complexity: 1) strictly asymmetric, 2) mixed symmetric/asymmetric, and 3) a two-phase model. Obviously, more complex model can fit the data better. However, can you show that the more complicated models are indeed necessary using some model selection techniques? Especially model 3) might be hard to justify: Looking at Figure 2, one cannot really observe two different phases and model 2) seems to do fairly well already. Here I strongly suggest to do Bayesian model selection e.g. using Bayes factor – recently it has been shown that this can be efficiently calculated e.g. using thermodynamic integration also in the context of biological dynamical systems.

In your model, cells are distinguished according to the number of times they have divided (the states *i*). In the experiment you can only observe telomere length for a cell, not number of divisions.

How are those two quantities related in your model (what's your observation/error model for the data)? This seems nontrivial, e.g. because a certain state *i* in the model doesn't correspond to a fixed observed telomere length (you mention that each division yield a loss of ~30-50bp) and this uncertainty will propagate with each division. Hence, based on a measured telomere length inferring the number of divisions is probably not unique (e.g. what about a measurement of telomere length = 55? It could have divided once, losing 55bp at once. Or it might have divided twice losing a bit less then 30bp each time). Please discuss these non-uniqueness using either Bayesian credible intervals or more so identifiability analysis using profile likelihood/posterior methods.

According to the stochastic model, you implicitly assume that cell-cycle times (in your case the waiting times in state *i*) are distributed exponentially (which they are clearly not in the real stem cell system). Please discuss if/how this assumption impacts on the results. Alternatively people nowadays often use delayed models by having the cell go through multiple states before actually dividing – then the Poisson model would be replaced by a log normal in the limit I believe. Here comparison to actual cell cycle distributions would be helpful.

For the section "Proliferation properties of stem cells during adolescence and adulthood" it remains unclear what data is actually fitted with this procedure. Are the same data shown in Figure 2? If so, it seems strange that you already discuss the fitted model in a previous paragraph and describe this 'Bayesian inference method' only later.

In the next section (“A single sample of telomere length distribution…”) you analyze the entire telomere distribution instead of only looking at the average. First, it is worth mentioning that these data correspond to the open symbols in Figure 2 (which of course shows only the mean again). Second, did you refit the model to those distributions? If so, how was this done?

Reviewer #3:

The authors develop a mathematical model in order to interpret data of telomere length shortening in hematopoietic stem cells from a relatively large number of patients. The mathematical model is used to infer dynamic properties of stem cell populations from data on telomere length distributions. Such analysis could quantify parameters of telomere dynamics from a single blood sample of a person. The model can be used as a tool to retrospectively infer stem cell dynamics, which could become useful in the personalized diagnosis and prognosis of disease.

I think that this is a very nice paper and of very high impact for the field. It is a show-case for how biologically realistic mathematical models can be coupled with clinical data in order to arrive at clinically useful insights, with potential to influence disease understanding and diagnosis. Therefore, I think that the quality of the paper is definitely suitable for a high-impact journal, such as *eLife*, and I recommend that this paper is published in the journal after some revisions.

I think that revisions could improve the paper further, and my suggestions are given as follows:

Following the Introduction, you have a section on "Modeling telomere length dynamics". I think that this could be improved. In particular, it is hard to get a detailed enough picture of the exact modeling approach from this section. Some of the details that I was missing in this section are explained in the Results section. So perhaps, all model description could be placed in a modeling section, and the Results section could just describe the results. This might make it easier to read the paper. Of course, a detailed and clear model description is given in the supplementary materials, so my comment is just about the readability of this particular section in the main text

I have a question regarding the model structure, and it might be useful to discuss this somewhere in the paper. With a certain probability, asymmetric division occurs, where stem cell division leads to one stem daughter cell and one differentiating cell. With the opposite probability, symmetric division was assumed to occur, where division gives rise to 2 stem cells. This is a biologically reasonable model. However, it is also possible to formulate this differently. For example, you can assume that with a certain probability, a stem cell divides to give rise to 2 stem daughter cells, and with the opposite probability, it divides to give rise to 2 differentiated daughter cells. In this way, you then probably would need to include feedback mechanisms to obtain stable and realistic dynamics. Would such a difference influence your results or not? I am not suggesting analyzing further models, but a discussion on model robustness might be useful for the paper.

Along similar thoughts, it could be helpful for the Discussion to include some text that discusses whether the reported results could change if different mathematical formulations or assumptions are used.

For clarity, the Abstract could point out more strongly that data have been collected for this and experimental procedures were used.

[Editors' note: further revisions were requested prior to acceptance, as described below.]

Thank you for resubmitting your work entitled "Reconstructing the in vivo dynamics of hematopoietic stem cells from telomere length distributions" for further consideration at *eLife*. Your revised article has been favorably evaluated by Naama Barkai (Senior Editor) and three reviewers, one of whom is a member of our Board of Reviewing Editors.. The manuscript has been significantly improved and the points of the referees have been taken into account. There is one technical comment of referee two that the authors should consider before acceptance, as outlined below. The new Figure 2—figure supplement 1 clarifying the difference between the two models is very useful and might be well suited as a main figure.

Reviewer #1:

The authors have carefully addressed the reviewer comments. The issues raised in my first report have been clarified and the text is now much clearer.

Reviewer #2:

In revised version of their manuscript "Reconstructing the in vivo dynamics of hematopoietic stem cells from telomere length distributions", the authors have incorporated the comments and suggestions of the three reviewers, increasing the quality of the manuscript considerably. Especially some confusion due to inconsistent nomenclature has been sorted out and makes the manuscript much easier to read.

In particular, the authors responded to all my concerns/issues and could sort out most of them. However, I would like to comment on some specific points and ask for additional clarification if possible.

Fitting procedure:

I have the feeling that it is a bit confusing/redundant now; to summarize this is what I found as fitting setup:

1) You fit your models in the section "In vivo measurements of telomere length…" giving some parameter estimates;

2) You perform ABC-rejection sampling in the next section, giving some idea about the posterior parameter distribution and uncertainties;

3) You do maximum likelihood estimates of the parameters (+ confidence intervals), again giving some idea about the uncertainties.

So, you fit the same data, with three different methods (actually I'm not sure what the difference between 1 and 3 is). This seems unnecessary and confusing. Why doing basically the same thing three times? Is this for different models, or for comparison? What’s the difference between 1 and 3?

One observes that the estimates a (slightly) different, e.g. telomere_loss_ABC = 0.071 vs telormere_loss_MLE=0.075. Is this difference due to the different "error measure", i.e. *R^[2]^* for ABC and quadratic distance/Gaussian error for MLE? Since you do ABC to get a handle on the uncertainties, you should report them (as you did for the MLE), i.e. the boundaries of the credibility intervals.

One major advantage of having the full posterior is that you can look at correlations of parameters. The one-dimensional confidence/credibility intervals (or an approximation) you can also obtain from MLE. Consider showing e.g. pairwise scatter-plots of the posterior samples to show whether there are some correlations/dependencies.

This whole subject discussed above should be straightened out in the final manuscript. Choose one suitable method and stick to it (I would suggest MCMC + Bayes factors, which gives you posterior uncertainties and model comparison or the ABC version). Otherwise the reader will get lost.

Model selection:

Thanks for incorporating some model selection. Any particular reason why you choose AIC and not BIC? Does the BIC select the same model? Since the multiphase model is inferior to model 2, you should be careful not to interpret this model too much.

Figure 2:

Thanks for showing the fits in Figure 2—figure supplement 1. This helps a lot to see why model 1 doesn't explain the data so well. Is there any reason not to put this figure into the main manuscript (instead of the old Figure 2 which doesn't show the linear fit)?

About my question on exponential waiting times:

My question was about cell cycle times, i.e. the time from a cell's "birth" (the division of its mother cell) until it divides. From what I understood when reading the supplement section "Connections to the Normal and Lognormal distribution", you look at the "distribution" *N^(i)^(t)*, i.e. the number of cells across states, which is Poisson (model 1) or generalized Poisson (model 2). These can then be approximated by normals/lognormals.

However, to me this is fundamentally different from a normal/lognormally distributed cell cycle time. The cell cycle time tells us when a single cell is going to divide. Your *N^(i)^(t)* tells us how a population of states spreads over the states *i*. I don't see an immediate connection between those to quantities. Please comment on that.

Reviewer #3:

The authors have addressed all of my comments and improved the manuscript accordingly. I think it should be accepted for publication in the current form.

---

## [Author Response]

Reviewer #1:

*Major points:*

*A basic problem I have when reading the manuscript is that the models of telomere length reduction address the telomere length distribution in the stem cell population. However the telomere length data is obtained for the cell population taken from blood or bone marrow that contains differentiated cells. So in order to analyze the data it appears that one would need a model for the telomere length distribution of those cells found in the samples. Why should this distribution correspond to the one of the stem cell pool described by the model? This is quite unclear.*

We agree with the reviewer. Ideally, we would like to measure the telomere length distribution within the stem cell population over time. Obviously this is infeasible in human populations. However, homeostasis in the hematopoietic system is maintained by rare stem cell differentiations, see for example the very recent papers (Busch et al., 2015; Sun et al., 2014). Non stem cells differentiate further and allow for the diversity of cells in the hematopoietic system. But these cells have a finite life time and are eventually washed out, see for example (Busch et al., 2015; Sun et al., 2014; Werner et al., 2011). Only stem cells are able to maintain homeostasis in the long run. Age dependent differences in telomere shortening across different lineages of haematopoiesis across individuals can thus only persist in the hematopoietic system, if they occur on the level of the maintained self-renewing cell population. But we agree, our inference would only be possible if the differentiation process affects telomeres in a way that does not change with age. To account for this assumption better, we have now extended our explanation in the manuscript.

*A related point is that the authors discuss very simple scenarios where only one step of differentiation from the stem cell pool is considered. In practice, differentiated blood cells could arise via several differentiation steps in a more complex scheme. This might change the telomere length distribution of the circulating cells. Such issues are not addressed in the paper.*

Please see our explanations above. The time dependent component of telomere shortening in homeostasis should ultimately arise from the stem cell compartment (the cell compartment that is responsible in maintaining homeostasis). We implicitly assume that the basic organization of the hierarchy remains approximately the same throughout life (it is also well conserved across mammals) and thus cannot be responsible for the persistent temporal changes in telomere length dynamics (of course there are many differentiation steps until all mature blood cells are generated, however this just results in a shift of the distribution towards shorter telomeres. This shift is assumed to be constant and not age dependent). Consequently, it suffices to analyse the population dynamics of the persistent time dependent component of the system to indirectly infer properties of the stem cell pool. We have now made these assumptions more explicit.

*A question arises from the statement in that "a cell loses typically 30 to 50kbp telomeric repeats per cell division" (subsection “In vivo measurements of telomere length suggest an increasing number of hematopietic stem cells during human adolescence”). This implies that the loss of repeats per division is itself a stochastic variable with mean and variance. In this case the distribution of telomere length differs from the distribution of cell states* i*. This is not discussed. Rather, the distinction is blurred by a confusing mix of language. For example in the subsection “Proliferation properties of stem cells differ during adolescence and adulthood”, the units of* r/N_0_
*are given as kbp/year even though by definition* r/N_0_
*is a rate with units 1 over time.*

We apologize for this imprecision. We describe telomere length on the level of cells as an average cell property. The dynamics of telomere length is defined on a population level, where telomere length changes by a cell division, but is accessed within a pool of other cells. The effect of two cell divisions with a small loss of telomeric repeats or one cell division with a larger loss of telomeric repeats is indistinguishable on the population level. We thus actually estimate a parameter r�cN0. Here Δc is the average loss of telomere repeats per cell division (for example measured in base pairs) and rN0 is the relative proliferation rate (measured in 1/time). It is this interconnection of telomere loss and cell turnover that hinders us to for example estimate the actual number of active hematopoietic stem cells with our approach. In the previous version, we implicitly absorbed the parameter Δc. But we agree this might be a source of confusion and now explicitly state this parameter where necessary.

The statement in referred to by the reviewer was meant to be an example. If we know the parameter Δc exactly, we can estimate the turn over rate of stem cells. This does not imply that the telomere shortening itself is a stochastic variable in our model. Biologically, the loss of repeats is almost certainly stochastic on the level of single telomeres. However, we consider the shortening of telomeres on the cell level. This corresponds to an average of 184 independent events, so we assume that the variability is small.

To avoid confusion, we have changed the former statement to: “If for example a cell looses on average 50 bp telomeric repeats per cell division…”.

We also extended our Discussion and explain that we use the average telomere length of a cell throughout the manuscript and therefore only consider the average shortening of telomere length per cell division.

*The presentation of the work is sometimes confusing because of imprecise language and confusing terminology. For example, in the subsection “The model predicts characteristic telomere length distributions for different ratios of symmetric and asymmetric stem cell divisions” you state: "Each stem cell proliferates randomly with a rate* r*". Again, in the subsection “Asymmetric cell divisions”: "*r *represents the proliferation rate of a cell". However looking carefully at Equation (S1) suggests that the parameter* r *is not what the authors say it is. The proliferation rate per cell rather appears to be given by the ratio* r/N_0_
*when using the terminology of the authors. Similarly, for the case with symmetric divisions, the proliferation rate per cell is* r/(rpt+N_0_). *This is consistent with the statement that the proliferation rate of a stem cell decreases with time but is inconsistent with the definition of* r *in the text.*

We apologize for this imprecision. If homeostasis would be driven by a single stem cell, this cell indeed has a proliferation rate r. In model 1 we assume that N0 stem cells drive homeostasis and consequently the contribution of each stem cell changes to an effective proliferation rate rN0 on the population level. In model 2, the number of stem cells changes over time and consequently, the effective proliferation rates of stem cells on the population level also change with age. We now explain this in more detail in the manuscript and point out important differences between model 1 and model 2. We also discuss the difference between the proliferation rate *r* and the effective proliferation rate of stem cells within a pool of stem cells.

*The statement "the assumption of strictly asymmetric divisions fails to explain the pronounced loss of telomere in infants" is unclear. What exactly is the evidence for a pronounced loss in infants? If one fits a linear law to the data in Figure 2 there would be a lot of variability at young age but not a "pronounced loss".*

We have now added a new supplement to Figure 2 that shows the linear and logarithmic decrease. It also depicts the average initial telomere length in new-borns. It illustrates that during adolescence, telomeres are lost at a faster rate compared to the loss of telomeres in adults.

We also added a model selection based on maximum likelihood methods, which are explained in the response to reviewer 2. These methods reject model 1 and favor model 2 as a potential explanation of the data.

*Related to this last point is the statement in the caption to Figure 2: "The decrease of the average telomere length slows down in children and becomes almost linear in adults." I do not understand how this statement follows from the data shown. The data does not appear to support this statement.*

We now added a new Figure (Figure 2—figure supplement 1) that shows both the logarithmic and linear best fit to the data in Figure 2. It also highlights the mean initial telomere length in new-borns. It shows that the differences between both curves are marginal in adults, but the linear curve underestimates the telomere length reductions in new-borns and during adolescence. In other words, the telomere length declines faster then predicted by the best linear fit during adolescence and is well approximated by a linear decrease during adulthood.

We also removed the reference to Figure 2 to avoid confusion here.

*The statement of fact that the "increased loss" of telomere length at young age is "an immediate consequence of an expanding stem cell population" is a strong statement, without a clear reasoning. It rather seems to be a suggestion based on the model that is here stated as plain fact. Also, the evidence for an "increased loss" is not compelling, as it is not easily seen in the data in Figure 2. I do not understand where this claim stems from. The statement about "increased loss… during adolescence" is unclear too. Which part of the data is meant here?*

We thank the reviewer for this important remark. We have added another supplement to Figure 2 that compares the linear and logarithmic model prediction. It emphasizes the increased loss of telomeres during adolescence, but the connection to an expanding stem cell population is of course indirect.

We have thus now rephrased our statement and now say:” Our model suggests that the increased loss of telomere length…”.

We also cite Figure 2—figure supplement 1 after the statement.

*The distinction of two phases discussed in Figure 4 is not very convincing and seems a bit arbitrary. Also, in Figure 4, it is unclear what data was analyzed with the Bayesian method. What problem is solved by introducing the distinction between two phases? The Bayesian approach also becomes conceptually puzzling when the probability p is now itself becoming distributed by a very broad probability distribution in phase 1 (Figure 4). What does this mean?*

We now discuss a model selection procedure based on a maximum likelihood (please see the detailed discussion in response to reviewer 2). Indeed, model 2 is favoured over a multiphase model with subsequent independent parameters.

*The linear fits to the data in Figure 6 are not really compelling as the data seems to be consistent with widely varying linear behaviors.*

We agree with the reviewer. The lines are only meant to represent a trend. We do not mean to claim that this increase or decrease is linear. We added an additional explanation to the caption of Figure 6 to emphasize this last point.

Reviewer #2:

*Major points:*

*The applied inference approach (termed 'Bayesian inference method') seems to be very simplistic. The authors essentially apply Approximate Bayesian Computation (ABC) rejection sampling (this term should be mentioned in the manuscript). These likelihood-free methods are usually applied when the likelihood function of the model is intractable. However, the authors show (e.g. S10, S20) that they can solve for the relevant quantities of their model. Hence a likelihood based inference approach is possible (e.g. maximum likelihood estimation + profile likelihoods, or full MCMC for posterior distributions) and should be used (in order to avoid the pitfalls associated with ABC). In fact, fitting S10 is just a linear regression problem.*

We thank the reviewer for this important comment. We now mention the full term, “Approximate Bayesian Computation rejection sampling”, in the main manuscript as well as in the supplemental information.

As the reviewer correctly stated, we are able to solve the dynamical equations of the model and thus can utilise methods beyond ABC. In fact, our “best” parameter estimations rely on standard fitting procedures using regression analysis (or just linear fitting in case of equation S10).

We revisited our data analysis and in addition now implemented standard likelihood estimates for our regression analysis. In the case of a linear regression they correspond to the minimal mean squared distance of course. The likelihood-based methods confirm our parameter estimates from standard regression analysis. This is now stated in the manuscript too.

*You consider three different models of increasing complexity: 1) strictly asymmetric, 2) mixed symmetric/asymmetric, and 3) a two-phase model. Obviously, more complex model can fit the data better. However, can you show that the more complicated models are indeed necessary using some model selection techniques? Especially model 3) might be hard to justify: Looking at Figure 2, one cannot really observe two different phases and model 2) seems to do fairly well already. Here I strongly suggest to do Bayesian model selection e.g. using Bayes factor – recently it has been shown that this can be efficiently calculated e.g. using thermodynamic integration also in the context of biological dynamical systems.*

We thank the reviewer for this very helpful suggestion. We performed a model selection analysis based on the Akaike information criterion (AIC), as well as a Bayesian information criterion, using our likelihood estimates.

Model 1 scores with a AIC of 2550, model 2 has an AIC of 2328 and a multiphase model with a minimum of 7 parameters yields an AIC of 2361 and therefore model 2 minimizes the AIC. Based on these values and according to established standard procedures, model 1 is only with probability p≈10−48 more likely to minimize the information loss compared to model 2. A multiphase model performs slightly better compared to the linear model 1, but still it is more likely to minimize information loss compared to model 2 with a probability of p≈10−8.

Thus, as also suspected by the reviewer, model 2 is highly favoured and both model 1 as well as a multiphase model with different parameters for each subsequent phase can be rejected as more likely explanations for the data presented in Figure 2.

We added this point to the manuscript and now discuss and score the models according to the model selection criteria.

*In your model, cells are distinguished according to the number of times they have divided (the states* i*). In the experiment you can only observe telomere length for a cell, not number of divisions.*

*How are those two quantities related in your model (what's your observation/error model for the data)? This seems nontrivial, e.g. because a certain state* i *in the model doesn't correspond to a fixed observed telomere length (you mention that each division yield a loss of ~30-50bp) and this uncertainty will propagate with each division. Hence, based on a measured telomere length inferring the number of divisions is probably not unique (e.g. what about a measurement of telomere length = 55? It could have divided once, losing 55bp at once. Or it might have divided twice losing a bit less then 30bp each time). Please discuss these non-uniqueness using either Bayesian credible intervals or more so identifiability analysis using profile likelihood/posterior methods.*

This is an important point and was also raised by reviewer 1. We kindly refer the reviewer to our reply to reviewer 1.

*According to the stochastic model, you implicitly assume that cell-cycle times (in your case the waiting times in state* i*) are distributed exponentially (which they are clearly not in the real stem cell system). Please discuss if/how this assumption impacts on the results. Alternatively people nowadays often use delayed models by having the cell go through multiple states before actually dividing – then the Poisson model would be replaced by a log normal in the limit I believe. Here comparison to actual cell cycle distributions would be helpful.*

We thank the reviewer for interesting remark. Indeed, in Model 1, where we assume a strictly constant population size, cell cycle times are distributed exponentially. Although stem cell divisions might be stochastic, we agree that this assumption is unlikely met in highly regulated tissue environments under the presence of feedback. Interestingly, Model 1 is also unable to describe the full spectrum of telomere length dynamics, as for example suggested by the model selection criteria.

However, we want to point the attention of the reviewer to Model 2, where we allow for an increasing stem cell population. In Model 2, we pick cells randomly for proliferation to allow for stochastic stem cell behaviour, but we also condition the system on a certain constant output of differentiated cells per unit of time. This condition has several consequences. First, it ensures that the output of the stem cell compartment is not steadily increasing with an increasing stem cell pool size, which would lead to very unrealistic scenarios, particularly in adults, where we can assume that the output of the stem cell compartment is approximately at a constant level during homeostasis. In a sense, this condition describes a feedback of the hematopoietic system on the regulation of stem cell proliferation. This leads to a continuous decrease of the proliferation rate of individual stem cells. As the number of stem cells increase linearly in time, this decrease is proportional to 1/age of healthy individuals during adolescence. This basic property is found in humans and in mice, see for example Rozhok and DeGregori, 2015, and Bowie et al., 2006.

However, this condition has another interesting consequence. As the reviewer correctly observed, Model 1 leads to a Poisson distribution, which in our case is also well approximated by a Normal distribution. Thus the variable of effective proliferation rtN0 is approximately normally distributed. Except for normalization, a transformation of variables x→ertN0, allows to transfer Model 2 into Model 1, where x=rptN0+1 is the transformed effective proliferation rate of Model 2. Thus, Model 2 results approximately in a Log-Normal distribution, and recapitulates what the reviewer suggested to be a consequence of a model with multiple states of cell cycles and is a natural component of our model without explicitly introducing such multiple states. It is tempting to speculate that the mechanism of multiple cell cycle states evolved to ensure such a regulated constant cell output in homeostasis.

We added a subsection to the supplemental information that discusses the connections of our Model 1 and Model 2 to the Normal and Log-Normal distribution in more detail, see for example equation (S26).

We also discuss this connection in more detail in the main text.

For the section "Proliferation properties of stem cells during adolescence and adulthood" it remains unclear what data is actually fitted with this procedure. Are the same data shown in Figure 2? If so, it seems strange that you already discuss the fitted model in a previous paragraph and describe this 'Bayesian inference method' only later.

We apologize for this imprecision. We perform the Bayesian inference method on the same data set that leads to Figure 2 and indeed this data was also used to fit the model in the previous paragraph. While we discuss the “best” fit of Model 1 and Model 2 to the data in Figure 2, we aim to discuss the range of parameters that are consistent with the model within a certain statistical measure in the next subsection. We explain this now in more detail and state more precisely which data is fitted, as well as the connection to the former section.

*In the next section (“A single sample of telomere length distribution…”) you analyze the entire telomere distribution instead of only looking at the average. First, it is worth mentioning that these data correspond to the open symbols in Figure 2 (which of course shows only the mean again). Second, did you refit the model to those distributions? If so, how was this done?*

Thanks for this comment; we stated in the caption of Figure 2 that open symbols correspond to cases where we have the full telomere length distribution of individuals (in total 66 healthy persons). This is now also mentioned in the main text.

The fitting procedure is described in the supplemental information section, see for example Equation (S29), which is now cited in the main text. The results of the fits are presented in Figure 6—figure supplement 1–Figure 6—figure supplement 3, and all parameters are summarised in [Supplementary-material SD1-data].

Reviewer #3:

*Following the Introduction, you have a section on "Modeling telomere length dynamics". I think that this could be improved. In particular, it is hard to get a detailed enough picture of the exact modeling approach from this section. Some of the details that I was missing in this section are explained in the Results section. So perhaps, all model description could be placed in a modeling section, and the Results section could just describe the results. This might make it easier to read the paper. Of course, a detailed and clear model description is given in the supplementary materials, so my comment is just about the readability of this particular section in the main text*

We thank the reviewer for this suggestion. We restructured the manuscript as suggested. We moved all model descriptions from the Results into the modelling section. In addition we cite some equations from the supplementary information to improve readability further.

*I have a question regarding the model structure, and it might be useful to discuss this somewhere in the paper. With a certain probability, asymmetric division occurs, where stem cell division leads to one stem daughter cell and one differentiating cell. With the opposite probability, symmetric division was assumed to occur, where division gives rise to 2 stem cells. This is a biologically reasonable model. However, it is also possible to formulate this differently. For example, you can assume that with a certain probability, a stem cell divides to give rise to 2 stem daughter cells, and with the opposite probability, it divides to give rise to 2 differentiated daughter cells. In this way, you then probably would need to include feedback mechanisms to obtain stable and realistic dynamics. Would such a difference influence your results or not? I am not suggesting analyzing further models, but a discussion on model robustness might be useful for the paper.*

We agree with the reviewer that there are different ways to formulate the model. The alternative proposal is certainly possible and reasonable, in fact some of the authors have worked on such models before (Dingli D, Traulsen A and Michor F. (A)Symmetric Stem Cell Replication and Cancer. PLoS Comput Biol. 2007;3(3):e53). The difference of both formulations will depend on the detailed assumptions of stem cell proliferation. If we assume a feedback driven successive proliferation of stem cells, both models result in the same time dependent properties of telomere length, as two asymmetric divisions have the same impact on the stem cell population as one symmetric renewal followed by a symmetric differentiation. Thus both situations would on average be indistinguishable on the level of telomeres. Only the interpretation of *p*, the probability of symmetric self-renewal would change.

In contrast to the average that is expected to be the same for both scenarios, the variance of the distribution might change. We would expect a higher variance in the case of symmetric differentiation and symmetric self-renewal. However, most likely this effect is small compared to the measurement related noise.

However, other properties such as the clonal load of hematopoietic cells, could change (Shahriyari and Komarova, 2013). Very recent studies of clonal dynamics in hematopoietic cells in mice seem to point to a model where stem cells contribute repeatedly to homeostasis with longer breaks of quiescence in between. This would suggest a system of predominantly asymmetric divisions rather a then a system of symmetric renewals and differentiations. However, this also might differ across different tissues (Morrison and Kimble, 2006).

We discuss this point now at the end of the manuscript.

*Along similar thoughts, it could be helpful for the Discussion to include some text that discusses whether the reported results could change if different mathematical formulations or assumptions are used.*

We kindly refer the reviewer to our explanation above. We added further discussion at the end of the manuscript.

*For clarity, the Abstract could point out more strongly that data have been collected for this and experimental procedures were used.*

Thanks for the remark. We now have an additional sentence in the Abstract to emphasize the collection and analysis of data in combination with the mathematical model.

[Editors' note: further revisions were requested prior to acceptance, as described below.]

Reviewer #2:

Fitting procedure: I have the feeling that it is a bit confusing/redundant now; to summarize this is what I found as fitting setup:

*1) You fit your models in the section "In vivo measurements of telomere length…" giving some parameter estimates;*

*2) You perform ABC-rejection sampling in the next section, giving some idea about the posterior parameter distribution and uncertainties; 3) You do maximum likelihood estimates of the parameters (+ confidence intervals), again giving some idea about the uncertainties.*

*So, you fit the same data, with three different methods (actually I'm not sure what the difference between 1 and 3 is). This seems unnecessary and confusing.*

We thank the reviewer to point out this shortcoming. The reviewer’s summary of the fitting procedure is correct.

We agree with the reviewer, in particular method 1 and 3 are so similar that they can be considered redundant now. We initially started with method 1 and included method 3 later to be able to do a proper model selection procedure. However, this introduced unnecessary potential for confusion and might even lead to misunderstandings. We therefore rearranged the manuscript. We now dropped method 1 and only discuss the parameter estimations in terms of maximum likelihood methods in the section: “In vivo measurements of telomere length suggest an increasing number of hematopietic stem cells during human adolescence”. Please see below some more detailed comments.

*Why doing basically the same thing three times? Is this for different models, or for comparison?*

We now dropped the first method from the manuscript to avoid repetition and confusion.

*What’s the difference between 1 and 3?*

Method one was based on a standard *R^[2]^*distance. Method 3 is a standard implementation of maximum likelihood inference for a regression analysis. One therefore assumes a normal distribution of errors and minimizes the quadratic distance between the model and the data. For a linear regression, this yields exactly the same estimators as the standard *R^[2]^*distance. For a non-linear regression, the estimates can differ slightly. However, we dropped method 1 and only use method 3 in our manuscript now, as this yields a straightforward way to implement a model selection procedure.

*One observes that the estimates a (slightly) different, e.g. telomere_loss_ABC = 0.071 vs telormere_loss_MLE=0.075. Is this difference due to the different "error measure", i.e.* R^[2]^
*for ABC and quadratic distance/Gaussian error for MLE?*

Our expertise on statistical inference is not exhaustive, but we also believe that the slight differences occur due to the different error measures (*R^[2]^*in ABC and Gaussian/quadratic distance for MLE). This is supported by the fact that a standard *R^[2]^*parameter estimate recapitulates the most likely ABC estimates and also the estimated error intervals are comparable.

*Since you do ABC to get a handle on the uncertainties, you should report them (as you did for the MLE), i.e. the boundaries of the credibility intervals.*

*One major advantage of having the full posterior is that you can look at correlations of parameters. The one-dimensional confidence/credibility intervals (or an approximation) you can also obtain from MLE. Consider showing e.g. pairwise scatter-plots of the posterior samples to show whether there are some correlations/dependencies.*

We thank the reviewer for this remark. We now also state the confidence intervals based on the ABC method.

*This whole subject discussed above should be straightened out in the final manuscript. Choose one suitable method and stick to it (I would suggest MCMC + Bayes factors, which gives you posterior uncertainties and model comparison or the ABC version). Otherwise the reader will get lost.*

We rearranged the presentation in the manuscript to avoid confusion and misunderstanding. We dropped method 1 and present the parameter estimation in the framework of maximum likelihood estimates. In addition we discuss the ABC more carefully and emphasize, that maximum likelihood and ABC operate on the same data set.

*Model selection:*

Thanks for incorporating some model selection. Any particular reason why you choose AIC and not BIC? Does the BIC select the same model?

There was no particular reason to choose AIC over BIC. The BIC criterion selects the models in the same order. We state this in the manuscript now.

Since the multiphase model is inferior to model 2, you should be careful not to interpret this model too much.

We thank the reviewer for this remark. We rephrased these statements and emphasize the limitations of the two-phase model.

*Figure 2*:

*Thanks for showing the fits in Figure 2—figure supplement 1. This helps a lot to see why model 1 doesn't explain the data so well. Is there any reason not to put this figure into the main manuscript (instead of the old Figure 2 which doesn't show the linear fit)?*

We thank the reviewer and gladly take this suggestion. We changed Figure 2 accordingly and present Figure 2—figure supplement 1 in the main manuscript as a new Figure 3 now.

*About my question on exponential waiting times:*

*My question was about cell cycle times, i.e. the time from a cell's "birth" (the division of its mother cell) until it divides. From what I understood when reading the supplement section "Connections to the Normal and Lognormal distribution", you look at the "distribution"* N^(i)^(t), *i.e. the number of cells across states, which is Poisson (model 1) or generalized Poisson (model 2). These can then be approximated by normals/lognormals.*

*However, to me this is fundamentally different from a normal/lognormally distributed cell cycle time. The cell cycle time tells us when a single cell is going to divide. Your* N^(i)^(t) *tells us how a population of states spreads over the states* i*. I don't see an immediate connection between those to quantities. Please comment on that.*

We agree with the reviewer, in general our model discusses the distribution of the population size *N^(i)^(t),* which corresponds to the distribution of cells across different states *i*. However, in the supplemental chapter “Connections to the Normal and Lognormal distribution", we discuss the distribution of the dimensionless variable x=rt/N (x = proliferation rate per cell × time). We bravely interpret this variable as a random variable and show that model 2 leads to an approximately lognormal distribution for this variable. This random variable x corresponds to the rate of individual cell divisions within a certain time interval and thus can be interpreted (maybe it is too far-fetched) as an average waiting time for cell divisions within the cell population.

We now extend our explanation in the supplemental information to distinguish these two aspects better. However as this is not essential to our model and we are not entirely sure if this connection is helpful or confusing, we would be open to drop this aspect if the reviewer advises us to do so.